# Sustainable reference points for multispecies coral reef fisheries

Jessica Zamborain-Mason [1,2,3] ✉, Joshua E. Cinner [3], M. Aaron MacNeil [4], Nicholas A. J. Graham [5], Andrew S. Hoey [2,3], Maria Beger[6,7], Andrew J. Brooks [8], David J. Booth[9], Graham J. Edgar [10], David A. Feary[11], Sebastian C. A. Ferse [12,13], Alan M. Friedlander[14,15], Charlotte L. A. Gough [16], Alison L. Green[17], David Mouillot [3,18], Nicholas V. C. Polunin[19], Rick D. Stuart-Smith [10], Laurent Wantiez [20], Ivor D. Williams[21], Shaun K. Wilson [22,23] & Sean R. Connolly [2,24]

Sustainably managing fisheries requires regular and reliable evaluation of stock status. However, most multispecies reef fisheries around the globe tend to lack research and monitoring capacity, preventing the estimation of sustainable reference points against which stocks can be assessed. Here, combining fish biomass data for >2000 coral reefs, we estimate site-specific sustainable reference points for coral reef fisheries and use these and available catch estimates to assess the status of global coral reef fish stocks. We reveal that >50% of sites and jurisdictions with available information have stocks of conservation concern, having failed at least one fisheries sustainability benchmark. We quantify the trade-offs between biodiversity, fish length, and ecosystem functions relative to key benchmarks and highlight the ecological benefits of increasing sustainability. Our approach yields multispecies sustainable reference points for coral reef fisheries using environmental conditions, a promising means for enhancing the sustainability of the world's coral reef fisheries.

In contrast to many industrial fisheries, where knowledge about stock status has informed rebuilding efforts and management for sustainability[1–4], multispecies coral reef fisheries are overwhelmingly data-poor[2]. However, given their importance to coastal people[5] and increasing anthropogenic pressures[6], it is critical for reef fisheries to be assessed if they are to be sustainably managed[7]. Assessing reef fisheries requires clearly defined reference points that can be linked to the best available estimates of stock size and catch data[8]. However, to date, such links have only been made at local scales and in a small number of places[9]. Relatively poor research and monitoring capacity in most regions where multispecies reef fisheries operate[2] have led to a lack of reliable long-term fishery information, preventing the estimation of location-specific sustainable reference points such as multispecies maximum sustainable yield (MMSY) and the standing stock biomass at which MMSY is reached ($B_{MMSY}$), and impeding the assessment of reef fish stocks at global scales[1].

Here, we estimate sustainable reference points for multispecies coral reef fisheries and provide a global assessment of the status of coral reef fisheries. Specifically, we (1) estimate these key MMSY and $B_{MMSY}$ fishery reference points for coral reef fish based on local environmental conditions; (2) provide a global assessment of the sustainability of multispecies coral reef fisheries from a long-term production perspective using available estimates of reef fish biomass and total catch (i.e., landings); and (3) highlight key ecological trade-offs between fisheries production and other indicators of ecosystem state.

## Results and discussion

### Sustainable reference points for multispecies reef fish assemblages

To establish MMSY and $B_{MMSY}$ reference points for coral reef fishes (Supplementary Table 1), we explored a range of common surplus production curves (e.g., Gompertz-Fox[10,11], Graham-Schaefer[12,13] and other versions of the Pella-Tomlinson[14]; Methods). The Gompertz-Fox surplus production model was favored in terms of predictive accuracy (Supplementary Table 2) so we focus principally on that model throughout this manuscript, but we also summarize results generated using other surplus production models in Table 1. Reference points were jointly estimated from the biomass trajectory of high compliance marine reserves ($n = 70$) and biomass in remote uninhabited reefs (defined as > 20 h away from human settlements; $n = 80$[15]). However, in contrast to previous fisheries-independent work aiming to estimate baselines and yields for coral reef fishes[16,17] (Methods), we estimate location-specific reference points based on local environmental conditions (i.e., reference points are estimated as explicit functions of sea surface temperature, ocean productivity, hard coral cover and whether the reef is an atoll). Additionally, we use gravity, a measure of the human population pressure present at a location[18], as a covariate to allow for the possibility that reserves may have depressed recovery trajectories[19–21] if they are embedded within fished seascapes of high human impact (i.e., if adult biomass is depleted outside of reserves, then import of biomass from nearby exploited areas will be reduced, leading to greater net export and thus lower potential biomass in reserves[21]; Supplementary Fig. 1; Methods).

We found that estimated MMSY and $B_{MMSY}$ for coral reef fish under average environmental conditions according to the Gompertz-Fox surplus production model were 5.8 [3.8–12.3] t/km²/y and 42.5 [35.9–151.7] t/km² (median [90% uncertainty intervals]; Supplementary Fig. 2), but site-specific estimates varied by almost an order of magnitude due to estimated differences in local conditions (posterior medians from 2.5 to 20.6 t/km²/y and 18.0 to 151.0

t/km², respectively; Fig. 1a, b). Expected MMSY and $B_{MMSY}$ were higher for atolls, for reefs with high coral cover and high ocean productivity, and they were expected to be lower in areas with high sea surface temperatures (Fig. 1c–h; Supplementary Fig. 3). Together, this helps explain the variability in suggested reference points in previous local fisheries-dependent studies (e.g., from 6 to 20 t/km²/y[22]) and highlights the importance of accounting for local context when assigning fisheries reference points. It also illustrates how coral loss and increased sea surface temperatures from ongoing human-induced environmental change could impact the long-term food provisioning from multispecies reef fisheries. Estimates of $B_{MMSY}$ produced by surplus-production models less favored by model selection tended to be higher (Table 1), but relationships with environmental variables were consistent (Supplementary Discussion 1).

### Status of the world's coral reef fisheries

We next assessed the status of coral reef fisheries open to extraction (i.e., excluding the marine reserves and remote reefs used to estimate reference points; $n = 1903$; Supplementary Fig. 4) with respect to MMSY and $B_{MMSY}$ sustainable reference points. Based on available data, we performed analyses at two different scales: individual reef sites and jurisdictions (typically countries, states, or territories; Supplementary Table 3). At the reef site scale, we compared the estimated standing stock biomass (adjusted for methodological covariates) and available per-unit area reconstructed reef fish catch[23–26] relative to the site-specific $B_{MMSY}$ and MMSY reference points. At the jurisdiction scale, we grouped available expected biomass (also weighted by the proportion of marine protected areas in a jurisdiction; $n = 49$; Methods) and per-unit-area catch estimates ($n = 108$) and compared them to jurisdiction-specific sustainable reference points (Fig. 2a, b; Supplementary Fig. 5), assuming that our sampled reefs were representative of jurisdiction-level conditions (Methods). As different countries and international fisheries organizations categorize sustainable fisheries in

**Table 1 | MMSY reference points and assessment results under different surplus production models**

|  |  | Gompertz-Fox | Graham-Schaefer | P-T 3 | P-T 4 |
|---|---|---|---|---|---|
| MMSY for average environmental conditions and non-atolls (t/km²/y) |  | 5.8 | 5.5 | 5.7 | 5.5 |
| Range of site-specific MMSY (t/km²/y) |  | [2.5 – 20.6] | [2.4 - 19.8] | [2.4 – 21.8] | [2.3 - 26.1] |
| $B_{MMSY}$ for average environmental conditions and non-atolls (t/km²) |  | 42.5 | 56.9 | 65.4 | 70.9 |
| Range of site-specific $B_{MMSY}$ (t/km²) |  | [18.0 – 151.0] | [23.8 – 200.8] | [27.3 – 234.6] | [28.2 – 294.8] |
| *% of exploited sites* | B<$B_{MMSY}$ | 52 | 64 | 68 | 70 |
|  | C>MMSY | 44 | 44 | 44 | 44 |
|  | Conservation concern | 77 | 82 | 83 | 85 |
| *% of jurisdictions* | B<$B_{MMSY}$ | 51 | 63 | 63 | 65 |
|  | $B_{weighted}$<$B_{MMSY}$ | 37 | 47 | 55 | 59 |
|  | C>MMSY | 47 | 47 | 47 | 48 |
|  | Conservation concern | 54 | 59 | 67 | 71 |

Reference points and percentages are based on posterior medians. Note that Gompertz-Fox surplus production model (shaded in gray) was favored in terms of predictive accuracy (Supplementary Table 2). P-T refers to other versions of the Pella-Tomlinson model (Supplementary Discussion 1).

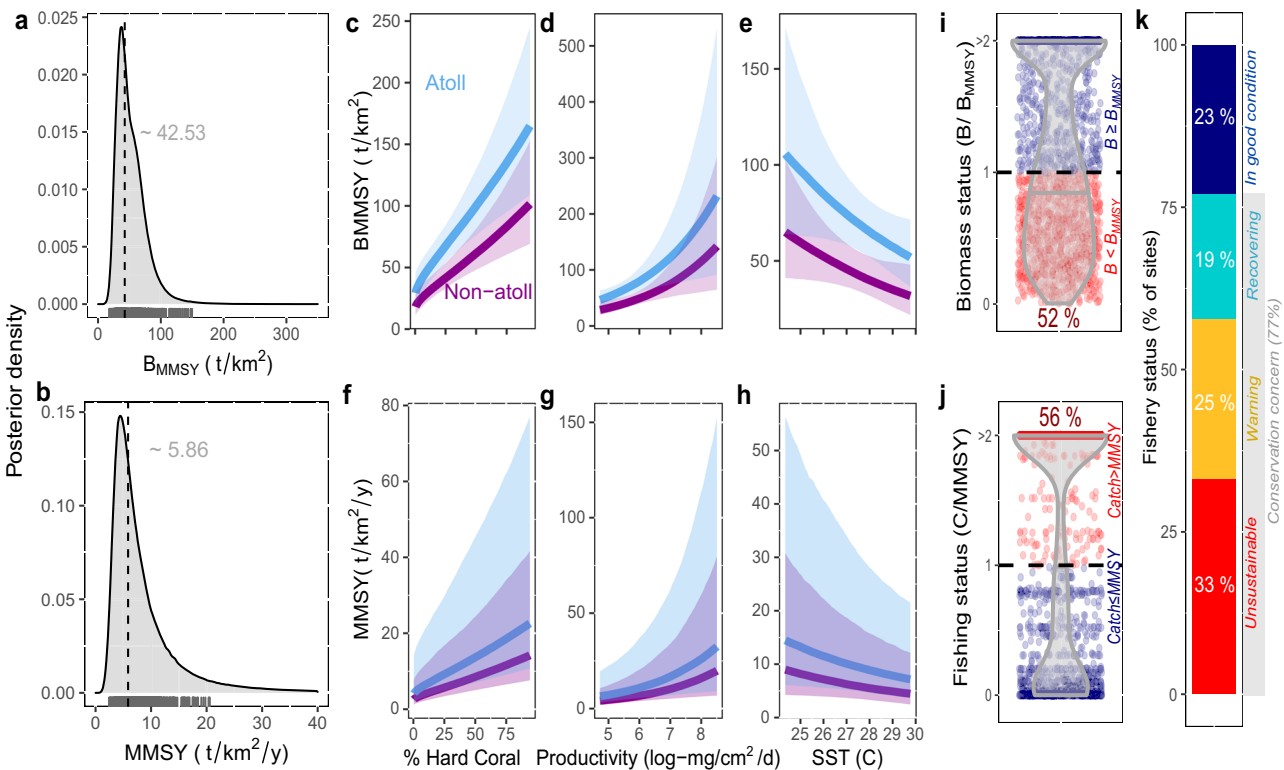

**Fig. 1 | Site-specific multispecies sustainable reference points and assessment for coral reef fisheries for the Gompertz-Fox surplus production model.** **a, b** Combined (i.e., inclusive of among-location variability and parameter uncertainty) site-specific MMSY (multispecies maximum sustainable yield) and $B_{MMSY}$ (biomass that produces multispecies maximum sustainable yield) posterior distributions. Rug plots show the posterior medians for each site given their specific environmental conditions ($n = 2053$ individual sites). Dashed lines and gray numbers represent the median posterior MMSY and $B_{MMSY}$ for average environmental conditions, respectively. **c–h** Expected change in MMSY and $B_{MMSY}$ for coral reef fishes with environmental conditions (hard coral cover, ocean productivity, sea surface temperature and whether the reef is an atoll). Line is the posterior median and polygons are 90% uncertainty intervals for atoll and non-atoll reef locations, with all other environmental variables fixed at their average values. See

Supplementary Fig. 3 for more details. **i, j)** Median biomass status ($B/B_{MMSY}$) and fishing status ($C/MMSY$) for each site open to extraction ($n = 1903$ individual sites). Jittered points are each site, color coded by (**i**) whether the estimated biomass (B) was above or below site-specific $B_{MMSY}$ (median ($B/B_{MMSY}$) < 1, red), and (**j**) whether the estimated per-unit-area catch (C) was above or below site-specific MMSY (median ($C/MMSY$) > 1, red). Numbers indicate the percentage of sites in each category that were below $B_{MMSY}$ (**i**) or estimated to be catching above MMSY (**j**). **k** Percentage of exploited sites assigned to different fishery status categories based on site-specific catch estimates, median biomass and surplus production curves: red (unsustainable), yellow (warning), turquoise (recovering), and navy blue (in good condition). Sites that have passed one of both reference points (i.e., MMSY and/or $B_{MMSY}$) are classified as being of conservation concern[4] (Methods). Source data are provided as a Supplementary Data file.

different ways[27,28], we took a production perspective and characterized a location based on whether it was below or above maximum production reference points ($B_{MMSY}$ or MMSY[28,29];). However, we also provide details with respect to "Pretty good multispecies yield" (PGMY) reference points, defined as the sustainable yield, and corresponding biomass range ($B_{PGMY}$), that is within 0.8 of MMSY[30]. For locations with both catch and biomass data available, we classified a location's sustainability status based on location-specific estimated surplus production curves (e.g., Fig. 2c).

Against site-specific MMSY benchmarks, we found that 52 [42–62]% of our sites open to extraction activities had median biomass values below their site-specific $B_{MMSY}$ from the Gompertz-Fox model (Fig. 1i) and 56 [49–63]% of sites had catch per-unit-area estimates above MMSY (Fig. 1j). Additionally, 65 [57–71]% of sites had catch levels indicative of overfishing (i.e., per-unit-area catch above the estimated surplus). A total of a total of 8 [5–12]% of sites had biomass values indicative of stock collapse (i.e., ≤0.1 of their estimated unfished biomass[8]), 23 % were below the lowest biomass value that produces PGMY, and less than half (47%) were in the biomass range of producing PGMY (i.e., estimated to be producing at least 80% of their maximum sustainable catch potential). Together, these results highlight that sustainable yields for more than half of reef sites open to extraction activities could potentially increase if stocks are allowed to recover and

catches span the range of available reef species. However, this would likely require a reduction of fishing pressure exerted on reefs.

At a jurisdiction scale, we found that 46 [28–60]% of 108 coral reef jurisdictions that had spatially reconstructed data were catching above MMSY (Fig. 2d; Supplementary Fig. 6). Additionally, 53 [33–63]% of the 49 jurisdictions with standing stock estimates had median biomass values in their exploited reefs below $B_{MMSY}$. The percentage of jurisdictions classified as below $B_{MMSY}$ decreased to 37 [23–47] % if we optimistically assumed that the proportion of waters protected within a jurisdiction were at unfished biomass conditions (i.e., calculating a weighted median biomass, Methods); and of those, 17 [17–26]% had biomass values indicative of fishery collapse. We found that 12 [12–14]% of jurisdictions had weighted biomass values below the lowest biomass value that produces PGMY and only 51 [41–55]% had weighted stock sizes within the range of providing PGMY. Note though that, given the shape of the surplus production curve (e.g., Fig. 2c), a higher percentage of jurisdictions are expected to be overfishing (i.e., catching above the surplus production) than those reported here as catching above MMSY if their standing stock values are not specifically at $B_{MMSY}$. For example, 41 [24–61]% of the 49 jurisdictions with both weighted biomass and catch estimates were overfishing their coral reef fish stocks based on available catch statistics.

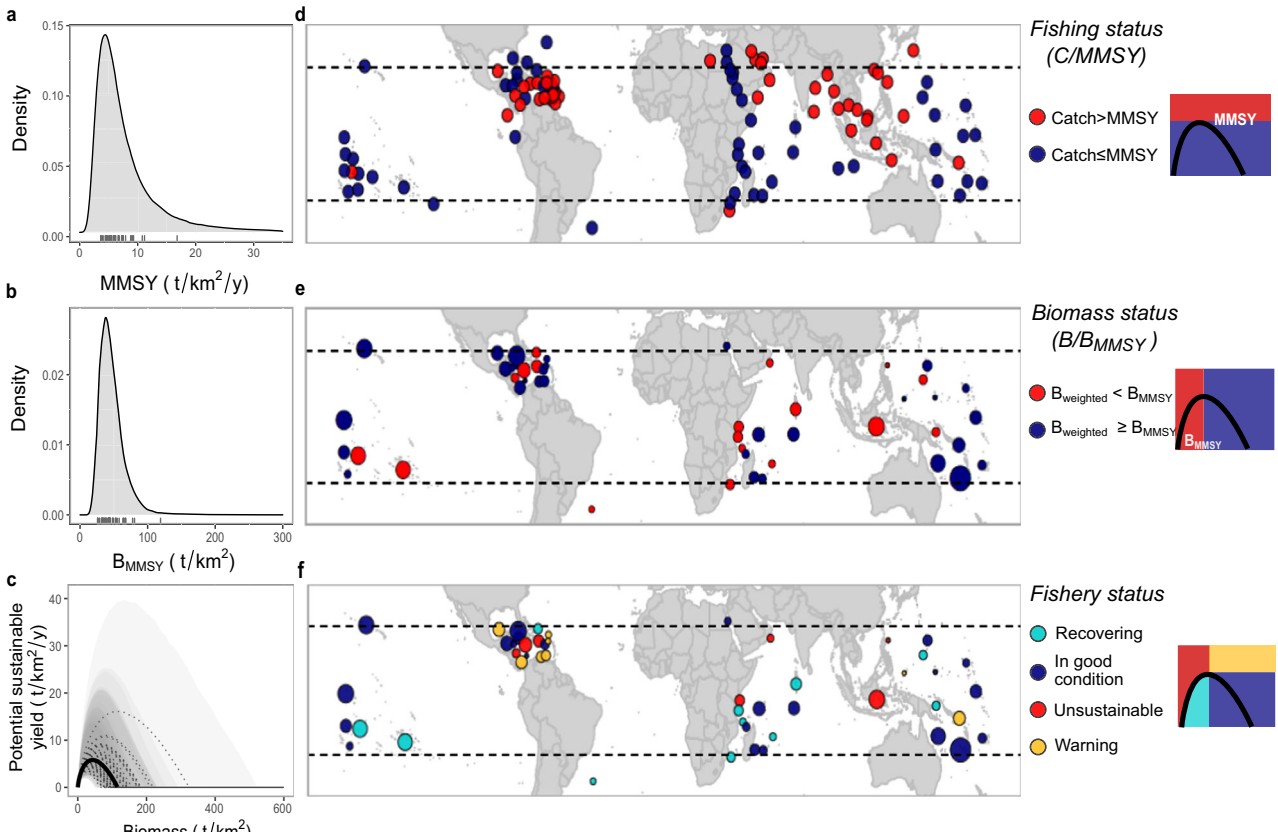

**Fig. 2 | Jurisdiction-level sustainable reference points and jurisdiction-level assessment of exploited reef fish stocks based on available information for the Gompertz-Fox surplus production model. a, b** Combined jurisdiction-level posterior distribution reference points (Methods; Supplementary Fig. 5). Rug plots are the medians for each jurisdiction with biomass data available (**n** = individual jurisdictions). **c** Median (dashed line) with 90% uncertainty interval (polygons) surplus production curve for each jurisdiction based on a jurisdiction's unfished biomass distribution and the posterior community growth rate. Darker shading indicates overlap of a larger number of jurisdiction-specific uncertainty intervals. Solid black line is the median surplus for average environmental conditions and dashed black lines are the median surplus for each jurisdiction. **d** Median jurisdiction fishing status (mean total catch (C; tonnes/km²/y) divided by jurisdiction-specific median MMSY ($n = 108$ individual jurisdictions). **e** Jurisdiction biomass status (median weighted biomass (B; tonnes/km²) divided by a jurisdiction's median $B_{MMSY}$; $n = 49$ individual jurisdictions. **f** Fishery status based on jurisdiction-specific catch, median biomass and surplus production curve estimates ($n = 49$ individual jurisdictions) color-coded by category: red (unsustainable), yellow (warning), turquoise (recovering), and navy blue (in good condition). Bubble size in (**e**) and (**f**) is scaled according to the number of sampled sites in each jurisdiction for which biomass values were recorded (ranging from 1 to 263). Diagrams to the right represent the categories based on total catch (y axes) and/or standing stock biomass (x axes). See Supplementary Figs. 5, 6 to see jurisdiction-specific reference point and status distributions (that show uncertainty for each jurisdiction). Source data are provided as a Supplementary Data file.

Combining both standing stock biomass and catch estimates, we found that 77 [68–85]% of exploited sites (n = 1903) with both standing stock biomass and spatial catch data available, or 53 [38–53]% of jurisdictions (n = 49) based on weighted biomass values, had reef fisheries of "conservation concern"[4], failing one or both sustainable reference points (i.e., C > MMSY and/or B < $B_{MMSY}$; Fig. 1k; 2e, f); 23 [15–32]% of sites or 46 [46–63]% of jurisdictions were *in good condition*, satisfying both sustainability benchmarks (C < MMSY; B > $B_{MMSY}$); 19 [17–21]% of sites or 23 [14–23]% of jurisdictions were likely *recovering*, having depleted biomass (B < $B_{MMSY}$) but reconstructed catch per unit area below the estimated production (C < surplus); and 25 [23–26]% of sites or 16 [6–16]% of jurisdictions were *warning*, catching above MMSY but with biomass above $B_{MMSY}$, meaning that stock biomass is expected to decline if current levels of fishing continue. When using the other surplus production models, all of which yield larger $B_{MMSY}$ reference points (Table 1), the percentage of locations classified as "conservation concern" increases from 77% to 85% for sites and from 53% to 71% for jurisdictions. Furthermore, assessment results did not improve substantially when we used reported catches instead of catch

reconstructions (Methods); with sites of "conservation concern" decreasing to 73% and jurisdictions to 42%.

## Trade-offs between long-term production and other ecosystem metrics

Maximizing production is not the only objective for ecosystem-based management aiming to sustain critical ecosystem states and processes[31,32]. By the time assemblage sustainable yields are met (e.g., MMSY), there are likely to be species that are overexploited and others that are not[1,28,33]. To evaluate the potential ecosystem impacts of fishing and trade-offs between production and ecosystem objectives, we examined how four ecosystem metrics (fish species richness[34], mean fish length, presence of top predators[35], and parrotfish scraping potential[36]) change along the surplus production curve (Fig. 3a; Supplementary Fig. 7; Methods). These analyses reveal the quantitative trade-offs between long-term production and ecosystem state, including the ecological costs of fishing unsustainably and the potential gains of increasing sustainability, assuming these ecosystem variables respond as expected to increases in community biomass.

Compared to reefs at unfished biomass, those at median MMSY values are expected to have, on average, lower total species richness (−18%), parrotfish scraping potential (−49%), mean fish length (−7%),

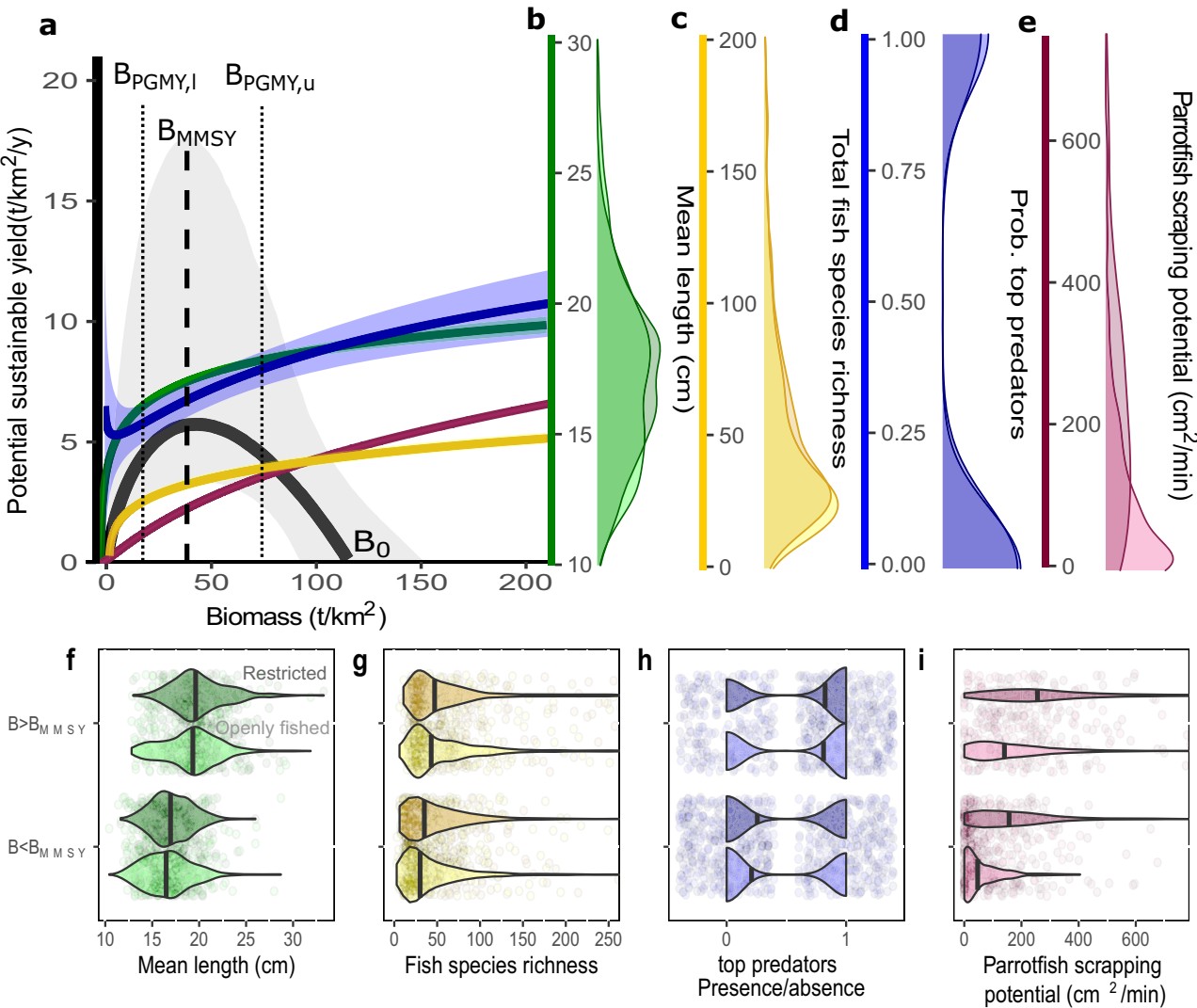

**Fig. 3 | Trade-offs among production and ecosystem metrics. a** Surplus production curve and expected values of ecosystem metrics as a function of biomass. Surplus production curve (black) is the posterior median (and 90% uncertainty intervals) sustainable yield for most common (for categorical variables) and average (for quantitative variables) sampling and environmental conditions. Ecosystem metrics are generalized additive model fits with 95% confidence intervals along the surplus production biomass gradient using metrics that are consistent with the surplus production curve conditions (marginalized for sampling and environmental covariates). Vertical lines represent the median biomass values at MMSY ($B_{MMSY}$) and pretty good multispecies yield ($B_{PGMY,l}$ is the lower bound and $B_{PGMY,U}$ is the upper bound) for average environmental conditions. Density distributions represent mean fish length (**b**; $n = 1763$ individual sites), total fish species richness

(**c**; $n = 1753$), presence/absence of top predators (**d**; $n = 1763$), and parrotfish scraping potential (**e**; $n = 1116$), of our sampled reefs open to extraction correcting for sampling effects. Note that (i) color scales in (**b**–**e**) are also the scales for the respective colors in (**a**), and (ii) "**a**" shows the probability of observing top-predators and "**e**" the density distribution of presence/absence of top predators. See Supplementary Fig. 7 for individual ecosystem metric relationships and distributions along the full range of biomass values. **f**–**i** Distribution of ecosystem metrics in reefs open to extraction separated as to whether the reefs were above or below site-specific $B_{MMSY}$ reference points. Jittered points are individual reef sites. In (**b**–**i**) dark colored density plots represent fished reefs with some level of gear or effort restrictions in place, and light-colored density plots represent openly fished reefs. Source data are provided as a Supplementary Data file.

and chance of encountering top predators (−20%; Fig. 3a, b). Further ecosystem losses (of −12, −24, −5, and −9 % respectively) are expected as biomass levels decrease to the lower bound of PGMY, where catches are 80% of MMSY and $B_{PGMY} < B_{MMSY}$, instead of MMSY. However, our analyses imply that going from $B_{MMSY}$ to the conservative side of PGMY (where $B_{PGMY} > B_{MMSY}$) would still maintain catches at 80% of MMSY and be associated with 9%, 23%, 4%, and 9% increases in these ecosystem metrics, respectively, relative to $B_{MMSY}$.

We found that exploited sites below $B_{MMSY}$ are performing worse for all ecosystem metrics in comparison to sites above their site-specific $B_{MMSY}$ reference point threshold (Fig. 3). However, separating our observed reefs into those that were openly fished and those that are fished but have active gear or effort restrictions in place highlights (i) the degree to which fisheries restrictions are

associated with enhanced production and ecosystem benefits[15], especially in terms of parrotfish scraping potential (median almost four times larger in restricted reefs compared to openly fished reefs; Fig. 3b–i; Supplementary Fig. 7), and (ii) that fisheries restrictions themselves might be insufficient to recover reef fisheries to maximum production values (i.e., many restricted sites were still below $B_{MMSY}$).

## Adapting reference points and assessments in the future

Given the limited availability of both catch and fishery-independent coral reef data, our study makes several assumptions that could be refined when updating sustainable reference points and assessing reef fisheries in the future. In this regard, we highlight five research avenues that our work suggests are likely to be particularly important for

improving estimates of sustainable benchmarks, particularly at the spatial scales most relevant for management.

The first of these is collecting data that increases our understanding of geographical variation in recovery trajectories. We used a space-for-time substitution among reserves of different ages to infer how biomass of reef fish grows with time[16,17], assuming the biomass at reserve age zero and the community growth rate of reef assemblages do not vary among locations. While time-series and space-for-time substitutions have been shown to give similar results for marine reserve biomass[37], time-series of multiple individual reserves could allow (i) estimates of reserve biomass starting points and community growth rates to vary among different locations, and (ii) additional inferences like reserve-specific export rates (Methods). Increased empirical recovery information may also help increase the accuracy of reference points (Methods) and better define the functional form of the surplus production curve for multispecies coral reef fish assemblages. We explored a range of alternative surplus production curves (e.g., Gompertz-Fox, Graham-Schaefer and other versions of the Pella-Tomlinson), and although the Gompertz-Fox was preferred in terms of predictive accuracy (Supplementary Table 2), all fit our empirical recovery data relatively well (Supplementary Discussion 1) with somewhat different implications for the estimated $B_{MMSY}$ reference point and the percentage of sites or jurisdictions classified as below $B_{MMSY}$.

A second priority is collecting relevant local-scale information to downscale our global results and increase their utility. As opposed to previous work (e.g., [15,16], we show that reference points can vary greatly among locations given their local context, and such variability can have materially different implications for local fisheries management. Here we provided fishery assessments at site and jurisdiction scales using available catch and biomass statistics, providing a global overview of the status of reef fish stocks. However, we acknowledge that uncertainty about stock status, catch statistics and their geolocation, and spatio-temporal heterogeneity within jurisdictions means that improved precision of estimates at the spatial and temporal scales appropriate to management are needed to better inform decisions by resource practitioners[38]. Our global model outputs can be combined with local-scale information (e.g., catch, biomass and reef area estimates) to provide baseline assessments at scales that match coral reef fisheries management in cases where local information alone would not be sufficient to estimate sustainability benchmarks. Additionally, collection of relevant environmental covariates at locations of concern where data are presently unavailable could also improve benchmark estimates. For example, we assumed average coral cover for sites without such data, yet obtaining this information, and additional metrics known to impact the biomass and productivity of reef fish (e.g., coral complexity[39] or non-reef associated habitats[40]), will help increase the accuracy of sustainable reference points and assessments.

Third, multispecies reference points and their functional dependence on environmental factors might need to be adapted if future reef systems transition to alternate stable states that differ substantially in species composition[41]. We do not know how individual species population-dynamic parameters (e.g., intrinsic growth rates) translate into community biomass-dynamic parameters (e.g., community biomass specific growth rates; Methods). Thus, it remains unknown how future reef assemblages, which could, for example, stabilize at different reef fish compositions, may alter reference points and their relationships with environmental conditions[42]. In this regard, we suggest that continuous monitoring of reef assemblages can help discern the life-history correlates of community long-term production (e.g., [43,44]), and thus help re-evaluate sustainable benchmarks for reef ecosystems that are shifting in response to ongoing environmental change.

Fourth, to understand what may be achieved through effective fisheries management and how to recover reef fish stocks, we need to find pathways that decouple the effect of fishing from other human-induced disturbances. Reliable fishing metrics (e.g., catch per unit effort) for most reef locations are absent, requiring the use of proxies like gravity[18] as a measure of local seascape human population pressure. However, gravity likely captures additional human impacts besides fishing pressure that can adversely affect fishery production. For example, reefs in areas of high human impact are expected to have lower biomass[32], are more likely to be below $B_{MMSY}$ reference points, but given their environmental conditions, in our analyses we found that they also tend to have lower reference point values (e.g., $B_{MMSY}$; Extended Data Fig. 8). Such interdependencies make teasing apart ecological capacity, fishing, and other human-induced disturbances analytically complicated (e.g., are we shifting baselines[45,46] by allowing high human impacted reefs to have lower reference points or do those regions have distinct ecological capacity?). Targeted research in regions with reliable catch statistics can be coupled with our model outputs to begin to disentangle fishing-mediated pathways versus other pathways by which metrics such as gravity impact both the status and potential dynamics of reef fisheries.

Finally, we believe that future multispecies reference points for coral reef fisheries would benefit from including a range of sustainability criteria beyond long-term production of the multispecies assemblage (including species winners and losers). In a similar way as our ecosystem metrics, there may be trade-offs between long-term yields and other desirable goals such as economic return[47] or nutritional yields[48], that will have to be evaluated to assess the overall sustainability of reef fisheries.

## Concluding remarks

Worldwide, coral reef ecosystems are experiencing widespread degradation in response to numerous anthropogenic threats[6]. While confronting the coral reef crisis requires international action on climate change[6], it is critical for reef fisheries to be managed sustainably so reef ecosystems can continue to provide food for millions of people and meet global sustainability goals[3]. Our study provides sustainable reference point estimates for coral reef fisheries based on environmental conditions that, combined with additional fishery information, allow an initial assessment for previously unassessed coral reef multispecies fisheries around the globe. Based on available data, our study estimates that most reef fish stocks open to extraction are currently compromised in comparison to reference points aimed at maximizing long-term production, and that important changes in ecosystem structure and function are associated with such assemblages, highlighting both ecological and production benefits of coral reef fish management and recovery.

## Methods

### Biomass and catch data

**Standing stock biomass.** Reef fish biomass estimates were recorded through underwater visual census (UVC) from surveys collected on a total of 2053 reefs spanning depths from 0 to 26 m and the following reef habitat types: slopes, crest, flat and lagoons/backreefs. Most sites came from our main dataset (e.g., [18]). However, additional sites from other published work[16,17] that used the same sampling methodology were also included. All surveys used standard belt-transects, distance sampling, or point-counts, and were conducted between 1999 and 2014. Except for the biomass trajectory of reserve reefs, where data from multiple years were available from a single reef, we included only data from the year closest to 2010. This was done because the majority of sites were only sampled once. Within each survey area, diurnally-active, non-cryptic reef fish above 10 cm length from families that are resident on the reef (Supplementary Table 1) were counted, identified to species level, and total length (TL) estimated, except for one data provider who measured biomass at the family level. Total observed biomass density of fish on each survey was calculated using published species-specific

length–weight relationships available on FishBase (http://fishbase.org[49]). When length–weight relationship parameters were not available for a species, we used the parameters for a closely related taxonomic level. Our selected reefs were originally classified into three different management groups: (i) openly fished (i.e., regularly fished without effective restrictions), (ii) restricted fishing—whether there were active restrictions on gears (e.g., bans on the use of nets, spearguns, or traps) or fishing effort (e.g., bag limits), and (iii) high compliance no-take marine reserves. However, for the purpose of our study some reefs were categorized as "remote" if they were uninhabited and more than 20 h away from human settlements[15]. We chose 20 h travel time because, for lower cut-off travel times (e.g., 10 h), biomass did not asymptote as a function of travel time (i.e., estimates of unfished biomass would be biased low when lower travel time thresholds are used; Supplementary Fig. 12). Thus, we ended up having four defined categories (i.e., remote, high-compliance marine reserves, restricted and openly fished): remote and high compliance marine sites ($n = 150$) were used to estimate the sustainable reference point parameters and fished sites (openly fished and restricted; $n = 1903$) were used to assess the status of individual reef sites open to extraction relative to reference points. See Supplementary Fig. 4 for a map of our sites.

**Reef fish catch.** Spatially reconstructed reef fish catch estimates (in metric tonnes) were obtained from the Sea Around Us Project (SAUP) catch database (http://www.seaaroundus.org[23,24]). We only used fish classified as "reef associated" species of the families included in our biomass estimates (Supplementary Table 1) from all sectors that intersected with coral reef polygons[25] and calculated the mean total catch per year for the period between 2008 and 2014. These spatial reconstructions record, for half degree spatial cells, an estimate of the catch obtained in a given year from each "fishing entity" (e.g., a country). We intersected this global spatial grid with global tropical coral reef polygons[25] to estimate the total reef fish catch per-unit-area (i.e., t/km²/y) per reef polygon, assuming that catches of reef-associated species of the families in our biomass data came from the coral reef habitat contained within that polygon. Next, to obtain site-specific catch-per-unit-area estimates, we intersected individual reef polygons with our individual sites. When individual reef sites did not overlap global reef polygons, we added a buffer and assigned the resulting catch per-unit-area. At the jurisdiction-scale, we calculated the catch per unit area (catch/km²/y) by dividing a jurisdiction's estimated mean total reef fish catch that overlapped with global reef polygons by the estimated total jurisdiction reef area[25]. We excluded from the analyses polygons shared by multiple jurisdictions (i.e., 0.3% of total spatial reef fish catch). Note that we used the mean catch because we only used seven points in time, highly correlated with each other (Pearson's correlation coefficient > 0.99). Catch reconstructions do not account for reef fish destined for the aquarium trade or non-commercially caught fish intended for the live reef fish trade[24]. In this respect, our status estimates are conservative (although the biomass of fish destined to the aquarium trade is likely a negligible contribution). We also performed a sensitivity analysis to the choice of catch statistics (see the *Sensitivity analyses and additional model checks* section below).

**Multispecies maximum sustainable yield reference points**
In contrast to previous fisheries-dependent reef fisheries studies[22,50–52], we used a fisheries-independent approach that treats the whole multispecies coral reef fish assemblage (Supplementary Table 1) as a single stock (i.e., an aggregate surplus production model[53]) to estimate sustainable reference points and assess the status of fished reef stocks (i.e., catch potential and/or ecological availability of the multispecies assemblage irrespective of method of capture and catchability).

Aggregate surplus production models, do not account for variability and differences in productivity among species in the species mix (i.e., losers and winners[1,29]). Nevertheless, they give a measure of system-level maximum yield[54] and are considered a better approximation of sustainable production for multispecies assemblages than single-species estimates[53,54]. For the multispecies assemblage, we evaluated a range of alternative surplus production models (see *Sensitivity analyses and additional model checks* below), finding that the Gompertz-Fox model[10,11] was best in terms of out of sample predictive accuracy based on our data (i.e., higher expected log predictive density and lower leave-out-one information criteria; Supplementary Table 2); we therefore used this model to estimate multispecies maximum sustainable reference points (MMSY and $B_{MMSY}$). Specifically, our model was:

$$P = \log(B_0)^* r^* B^* \left(1 - \left(\frac{\log(B)}{\log(B_0)}\right)\right) \tag{1}$$

$$B_{MMSY} = \frac{B_0}{e} \tag{2}$$

$$MMSY = \frac{r^* B_0}{e} \tag{3}$$

where $P$ is the potential yield or annual surplus production, $r$ is the community biomass specific growth rate (analogous to intrinsic growth rate in population growth; hereafter called community biomass growth rate), $B$ is the standing community biomass, $B_0$ is the unfished community biomass and $e$ is the euler number (i.e., 2.718281828). Note that the estimated community growth rate for the multispecies assemblage does not necessarily correspond to a weighted average of the individual species (see *Sensitivity analyses and additional model checks* section).

We analyzed the entire dataset (s) using different models for three subsets: reserves (i), remote (j) and fished (z) (i.e., s = i + j + z). However, different components informed distinct parameters.

Seascape-level unfished biomass and community biomass growth rate for coral reef fish were jointly estimated[17] from the biomass trajectory of high compliance marine reserve sites ($n = 70$) and the reef fish biomass of remote reefs ($n = 80$[15];). Data from almost all reserves consisted of only one or a few years of data, precluding estimation of the variability in recovery trajectories among reserves. Instead, for the reserve sub-model, a space-for-time substitution approach between previously-fished high-compliance reserve sites of different ages that had environmental information was used[16,17,37], assuming that the relationship between reserve age and standing biomass follows a common Gompertz-Fox recovery trajectory, accounting for the human impact (i.e., total gravity[18]) of the location (i.e., allowing for a lower recovery biomass if human impact was above zero, reflecting net movement of biomass from reserves to surrounding fished areas: Supplementary Fig. 1). Note this differs from previous studies, data from which are included in this study (Supplementary Discussion 2). To ensure comparable representation of reserves in the dataset, if a reserve was sampled multiple times (i.e., at different ages), we randomly chose one year and checked that the randomly selected years did not affect the robustness of our trajectory estimates (Supplementary Fig. 11). For the remote sub-model, reef scale biomass observations contributed to the estimated seascape unfished biomass, bounding the potential values that reserves could reach (Supplementary Fig. 1; see *Sensitivity analyses and additional model checks*).

Available reef-specific methodological and environmental covariates ($x_{,s}$) thought to influence standing biomass or reef productivity, as well as jurisdiction-specific random effects ($u_c$), were considered at corresponding components. These covariates fell into three classes.

Environmental covariates were net primary production (NPP[55,56]), sea surface temperature (SST[57]), average proportion of substrate occupied by hard coral cover, and whether the reef is an atoll. These were assumed to directly influence the carrying capacity of reef fish populations. Sampling covariates included census method (i.e., standard belt transect, point count), sampling area, habitat type (i.e., flat, crest, backreef), and depth of survey (m). Given that the scale of metapopulation closure for reef fish likely spans different habitat types[58], and that we do not have the proportion of habitat types for each population, we modeled habitat type as a methodological covariate, assuming that the distribution of habitat types is relatively consistent across populations (in which case a biomass estimate from a given habitat is a biased estimate of that location's biomass). We had two additional reef-specific covariates to correct for potential reserve biases: reserve size and gravity (size of human populations in the surrounding seascape divided by the accessibility- in minutes of travel time squared- of reef sites to them[18], the latter of which is a measure of human impact that we introduced to control for potential reserve exports (see *Sensitivity analyses and additional model checks*). Categorical covariates were treated as dummy variables (1's and 0 s), and continuous methodological, environmental, and reserve size covariates were standardized (mean-centered and divided by two standard deviations[59]). Gravity was not standardized (i.e., not mean centered) so that estimated baseline parameter values from the reserve recovery (e.g., MMSY and $B_{MMSY}$) would correspond to the values that they would be expected to have under average environmental conditions, and when human population pressure is zero.

We first assessed collinearity among our covariates using variance inflation factors and pairwise correlations. We did this for the entire dataset, but we also tested different subsets because different subsets informed different parameter estimates. Pairwise correlations and variance inflation factors for the entire dataset did not show any collinearity concerns (all pairwise correlations were below 0.6 and VIF were below 1.5). Subsetting the data revealed some interdependencies among covariates in some subsets. For example, in reserve sites, reserve size was correlated with sampling area such that larger reserves had larger sampling areas (Pearson's correlation = 0.56). In remote reefs, atolls had lower ocean productivity compared to non-atolls (Pearson's correlation = 0.96). These interdependencies did not impact model convergence, probably because effect sizes from covariates correlated in some subsets were informed by other subsets where correlations were much weaker.

Next, we tested the utility of two alternate models of varying complexity in capturing the structure of our data by following a "Principled Bayesian workflow"[60] on each model (Supplementary Methods). Models tested were as follows: a null model (which included just reserve age but no other covariates), and a full model (model which included all selected covariates and random effects in the fished component of the data). This workflow revealed that the full model (which included all covariates) provided non-biased and informative reference points (z-scores from different simulations scattered around zero and mean posterior contraction values > 0.5). Model selection through leave-out-one cross-validation, also favored the full model (Supplementary Table 2), indicating that the model including all covariates had better predictive accuracy[61], so we explain this best-fit model in more detail below.

Different sub-models informed partially-overlapping subsets of the model parameters. The reserve sub-model (biomass-dynamic model) informed the biomass at reserve age 0 (i.e., $B_{min}$), the effect sizes (β parameters) for the environmental, reserve size, human impact and sampling covariates, the seascape-scale community biomass growth rate (r), and unfished biomass ($B_0$). The remote sub-model informed estimates of unfished biomass and the effect sizes for environmental and sampling covariates. The remote data also indirectly informed the community biomass growth rate because remote

reefs bound the estimates of unfished biomass and influence the difference between unfished biomass and the asymptotic biomass in reserves (Supplementary Fig. 1). Lastly, the fished sub-model was used to marginalize biomass for sampling effects and estimate the status of fished reefs. This sub-model also informed effect size estimates of gravity and the sampling covariates. Note that environmental covariate values overlapped substantially among sub-model categories (Supplementary Fig. 4) indicating that our reference points informed by our reserve and remote sites would not be biased due to lack of overlap with fished reefs in the distribution of environmental conditions (see *Sensitivity analyses and additional model checks*). Specifically, our best-fit model was:

$$\log(B_i) \sim N(\mu_i, \sigma_i) \tag{4}$$

$$\log(B_j) \sim N(\mu_j, \sigma_j) \tag{5}$$

$$\log(B_z) \sim N(\mu_z, \sigma_z) \tag{6}$$

$$B_{0,i} = \exp^{\log(B_0) + \beta_1 x_{oceanprod,i} + \beta_2 x_{SST,i} + \beta_3 x_{atoll,i} + \beta_4 x_{coral,i}} \tag{7}$$

$$\mu_i = \log\left(B_{0,i} * \exp^{\log\left(\frac{B_{min}}{B_{0,i}}\right) * \exp^{-rt_i}}\right) + \beta_5 x_{depth,i} + \beta_6 x_{crest,i}$$
$$+ \beta_7 x_{lagoon/backreef,i} + \beta_8 x_{flat,i} + \beta_9 x_{pointcount,i}$$
$$+ \beta_{11} x_{samplingarea,i} + \beta_{12} x_{size,i} + \beta_{13} x_{grav,i} \tag{8}$$

$$B_{0,j} = \exp^{\log(B_0) + \beta_1 x_{oceanprod,j} + \beta_2 x_{SST,j} + \beta_3 x_{atoll,j} + \beta_4 x_{coral,j}} \tag{9}$$

$$\mu_j = \log(B_{0,j}) + \beta_5 x_{depth,j} + \beta_6 x_{crest,j} + \beta_7 x_{lagoon/backreef,j}$$
$$+ \beta_{10} x_{distancesampling,j} + \beta_{11} x_{samplingarea,j} \tag{10}$$

$$\mu_z = \gamma + \beta_5 x_{depth,z} + \beta_6 x_{crest,z} + \beta_7 x_{lagoon/backreef,z} + \beta_8 x_{flat,z}$$
$$+ \beta_9 x_{pointcount,z} + \beta_{10} x_{distancesampling,z} + \beta_{11} x_{samplingarea,z} + \beta_{13} x_{grav,z} + u_c \tag{11}$$

where $i$, $j$, and $z$ index reserves, remote reefs, and fished reefs, respectively. $B_i$ is the biomass of reserve $i$, $B_j$ the biomass of remote reef $j$, $B_z$ the biomass of fished reef $z$, $t_i$ is the age of reserve i, and $B_0$ is the unfished biomass for average and most common environmental and sampling conditions. $B_{min}$ is the estimated biomass at reserve age 0, $r$ is the estimated community biomass growth rate, which we assume are consistent among reserves, absent of the reef-scale effects, owing to the scarcity of global single-reserve recovery data (space-for-time substitution). $\beta_{(1-4)}$ are the jointly estimated linear slopes corresponding to the environmental covariates, $\beta_{(5-11)}$ are the jointly estimated linear slopes corresponding to the sampling covariates, $\beta_{12}$ is the effect of reserve size on log-biomass (only on reserve component), $\beta_{13}$ is the effect of human impact on log-biomass, $\gamma$ is the intercept of fished reefs, $\sigma_{(i-z)}$ are the estimated standard deviations for the residual among-site variation in log-biomass, and $u_c$ represents the random effects for jurisdiction $c$.

It is important to note that $B_0$ represents unfished biomass at the seascape scale (i.e., unfished biomass when fishing is negligible at the spatial scale of approximate population closure), as might be expected on remote reefs. In contrast, reserves are typically nested within fished seascapes where biomass tends to be depleted, so we would expect net export from reserves and thus a reserve biomass equilibrium somewhat below $B_0$ if human impact is above zero (Supplementary Fig. 1). Such effects would be incorporated in the human impact effect size

parameter: if human impact is estimated to be above zero, the estimated biomass would be expected to be adjusted in comparison to the observed biomass with equivalent environmental and sampling covariate values.

Model parameters were given the following priors:

$$\log(B_0) \sim N(\log(120),1) \tag{12}$$

$$\log(r) \sim N(-2,1) \tag{13}$$

$$\log(B_{\min}) \sim N(\log(10),1) \tag{14}$$

$$p \sim U(0,1) \tag{15}$$

$$\beta_{..} \sim N(0,2) \tag{16}$$

$$\sigma_{..} \sim Cauchy(0,1) \tag{17}$$

$$u_{..} \sim N(0,\sigma\_u) \tag{18}$$

$$\sigma u_{..} \sim Cauchy(0,1) \tag{19}$$

$$\gamma \sim N(5,5) \tag{20}$$

$B_0$, $r$, $\sigma$ and $B_{\min}$ were constrained to be non-negative. All scenarios were run using the Hamiltonian Monte Carlo algorithm implemented in RStan[62]. Four chains were run for each scenario, leaving 4000 samples in the posterior distribution of each parameter. Convergence was monitored by running four chains from different starting points, examining posterior chains and distribution for stability, checking that the potential scale reduction factor (also termed R_hat) was close to 1 (below 1.01) and examining the effective sample sizes (>400) and rank plots[63]. Identifiability was examined by inspecting posteriors vs. prior distributions and by calculating posterior contraction values[60]. All parameters had contraction values above 0.69 when fitted to our data. Model fit was examined by posterior predictive checks, checking residuals against fitted values and ensuring residuals were normally distributed around zero (Supplementary Fig. 9).

## Site-specific reference points and assessment

Assuming that sampled reefs are representative of the conditions at the scale of population closure, we assessed the biomass status of sites open to extraction (i.e., excluding marine reserves and remote reefs used to estimate sustainable reference points) by comparing a site's biomass (corrected for methodological effects) to the site-specific estimated $B_{MMSY}$ value. Site-specific reference points ($B_{MMSY}$ and MMSY) for every site in our data were estimated using the estimated posterior unfished biomass and community growth rate and our sites' available environmental information:

$$B_{0,s} = \exp^{\log(B_0) + \beta_1 x_{oceanprod,s} + \beta_2 x_{SST,s} + \beta_3 x_{atoll,s} + \beta_4 x_{coral,s}} \tag{21}$$

$$MMSY_s = (B_{0,s}{*}r)/e \tag{22}$$

$$B_{MMSYs} = B_{0,s}/e \tag{23}$$

(Note that for sites that did not have available coral cover information we assumed that coral cover was at its mean level in the database). To make biomass estimates comparable to reference points, standing stock observed biomass estimates were corrected for methodological covariates (Habitat type, Depth, Census method, and Sampling area) using the posterior effect sizes from the reference point model and calculating the marginalized biomass (i.e., corrected biomass as if it was collected for slopes, using standard belt transects, average depth and sampling area). Then for reefs open to extraction, we compared these site-specific marginalized biomass estimates ($B_{marg,z}$) and per-unit-are catch estimates ($C_z$) to their estimated $B_{MMSY}$ and MMSY values, defining a location as to whether its biomass status ($B_{status,z}$) or fishing status ($F_{status,z}$) were below or above/equal to 1. We report the median biomass status and 90% uncertainty intervals:

$$B_{marg,z} = \exp^{\log(B_z) - (\beta_5 x_{depth,z} + \beta_6 x_{crest,z} + \beta_7 x_{lagoonbackreef,z} + \beta_8 x_{flat,z} + \beta_9 x_{pointcount,z} + \beta_{10} x_{distancesampling,z} + \beta_{11} x_{samplingarea,z})} \tag{24}$$

$$B_{status,z} = B_{marg,z}/B_{MMSY,z} \tag{25}$$

$$F_{status,z} = C_z/MMSY_z \tag{26}$$

To estimate the relative catch potential for our sites, we also calculated the potential sustainable yield or surplus ($P_z$) of that site, conditional on its estimated biomass, and we expressed this relative to that site's estimated MMSY ($MMSY_z$), which is of course the catch potential of a site for the specific case when the estimated biomass of the site is equal to $B_{MMSY}$:

$$P_z = \log(B_{0,z}){*}r{*}B_{marg,z}{*}\left(1 - \left(\frac{\log\left(B_{marg,z}\right)}{\log\left(B_{0,z}\right)}\right)\right) \tag{27}$$

$$C_{pot,z} = P_z/MMSY_z \tag{28}$$

$C_{pot,z}$ would thus have a value of one if a location's biomass ($B_{marg,z}$) is at $B_{MMSY,z}$ and below one as the biomass is above or below $B_{MMSY,z}$.

We characterized sites' overfishing and fishery status based on site-specific estimated surplus production-curves. A site was categorized as subject to overfishing if total per-unit area catch ($C_z$) was above the estimated surplus ($P_z$). Additionally, sites were considered in "good condition" if biomass was above its site-specific $B_{MMSY}$ and the total catch was below the site's MMSY value; "unsustainable" if the site was simultaneously subject to catches above its estimated surplus production curve and had biomass estimates below $B_{MMSY}$; "warning" if biomass was above $B_{MMSY}$ but catches were above MMSY (i.e., on average expecting the stock to decline in the long-term); and "recovering" if the site had biomass values below $B_{MMSY}$ but not catching above its estimated surplus production curve (see schematic representation in Fig. 2f). Note we classify a stock as recovering based on fisheries productivity[28], but we acknowledge that fishing per-se might not be the only factor influencing whether a stock is recovering. Finally, similar to ref. 4, we classified locations as "conservation concern" when catch was above MMSY and/or biomass was below $B_{MMSY}$. Such "conservation concern" status does not imply risk of extinction or

expectation of collapse; instead, it implies fisheries management is likely needed to restrict catches and/or recover reef fish stocks to maximize long-term fisheries production.

## Jurisdiction-level reference points and assessment

To assess biomass status (i.e., biomass relative to $B_{MMSY}$) and level of fishing (i.e., catching above MMSY or not catching above MMSY) of reefs open to extraction at a jurisdiction scale, we used jurisdictions that had available reef fish catch (Sea Around Us project[23,24]), reef area[25,26], and/or biomass estimates. At the jurisdiction scale (c), we compared the catch and biomass to their jurisdiction-specific estimated multispecies maximum sustainable yield reference points ($B_{MMSY,c}$ and $MMSY_c$). Jurisdictions typically represent countries or states with individual Exclusive Economic Zones. However, due to the scale of reef area estimates, some Exclusive Economic Zones were aggregated for the analyses and correspond to single jurisdictions (Supplementary Table 3). For jurisdictions for which we had biomass information, we used the distribution of site-specific sustainable reference-points ($B_{MMSY,s}$; $MMSY_s$) per jurisdiction (Supplementary Fig. 5), and, keeping the 4000 samples from the posterior, used the average of these as the jurisdiction-specific reference points (e.g., if a jurisdiction had two sites, we averaged site-specific posterior samples to get the jurisdiction posterior):

$$B_{0,c} = \text{mean}(B_{0,s,c}) \tag{29}$$

$$MMSY_c = \text{mean}(MMSY_{s,c}) \tag{30}$$

$$B_{MMSY,c} = \text{mean}(B_{MMSY,s,c}) \tag{31}$$

$$P_c = \log(B_{0,c}) * r * B * \left(1 - \left(\frac{\log(B)}{\log(B_{0,c})}\right)\right) \tag{32}$$

Where $B_{0,c}$, $MMSY_c$, $B_{MMSY,c}$, and $P_c$ are the jurisdiction-specific distributions for unfished biomass, MMSY, $B_{MMSY}$ and surplus production along a gradient of biomass (B), respectively. Similarly, for jurisdictions without biomass information we used the combined jurisdiction MMSY distribution recognizing that MMSY is likely within MMSY estimates for all jurisdictions for which we do have biomass data (instead of using average environmental conditions).

In a similar way, using the entire posterior distribution for each site open to extraction, we calculated a jurisdiction's estimated biomass distribution:

$$B_c = \text{mean}(B_{marg,z,c}) \tag{33}$$

However, such approach does not capture potential biomass subsidies from reserves within a jurisdiction, so a jurisdiction's biomass was also weighted by the reported proportion of territorial waters in marine protected areas for the jurisdiction or parent jurisdiction (i.e., areas that have been reserved by law or other effective means to protect part or all of the enclosed environment[64]):

$$B_{weighted,c} = (\text{mean}(B_{marg,z,c}) * (1 - p_{mpa})) + (\text{mean}(B_{0,s,c}) * (p_{mpa})) \tag{34}$$

where $B_{marg,z,c}$ is the marginalized biomass of fished sites (z) and $B_{0s,c}$ is the unfished biomass of all sites for a given jurisdiction (c) and $p_{mpa}$ is the proportion of territorial waters protected. As we do not know the biomass of protected reefs in all jurisdictions, we took an optimistic approach, and assumed that the biomass in territorial waters protected for a given jurisdiction was equal to the estimated unfished biomass (acknowledging that this scenario is optimistic because most reserves will likely be below $B_0$ if they act as net exporters). It is this

optimistic scenario that is shown in our figure (Fig. 2), although eight jurisdictions (PRIA, Australia, Hawaii, Belize, Reunion, New Caledonia, Mexico and Northern Mariana Islands) changed status if we used only the reefs open to extraction (from above to below $B_{MMSY}$; Supplementary Fig. 6). Additionally, to test that our sampled standing stock biomass estimates were representative of their jurisdiction and not significantly biased towards more accessible reefs, we compared a jurisdiction's mean total gravity to the mean total gravity of our reefs in that jurisdiction[19]. We found no evidence that our sample locations were biased (i.e., 95% confidence intervals overlap the unity line; Supplementary Fig. 13).

Although sustainability can be defined in different ways[65], and different countries use different thresholds to define whether over-fishing is occurring or the stock is overfished[28,29], here, a jurisdiction's fishing status was defined as catching above the maximum that can be sustained if mean total catch/km²/y was above its jurisdiction-specific MMSY estimates (i.e., C/MMSY > 1). Using MMSY allowed us to clearly identify jurisdictions without biomass information that were overfishing[27]. However, note that, based on the shape of the assumed surplus production curve (Fig. 2c, f), a higher percentage of jurisdictions are likely to be overfishing than we have estimated as catching above MMSY if two conditions are met: (a) they have biomass values below the estimated $B_{MMSY}$ reference point, and (b) catch levels are between MMSY and the surplus production for that biomass. We used catch volumes instead of effort because (i) catch estimates are directly available from global catch databases (e.g., [23,24]), and for data-poor coral reef regions catch volume reporting tends to be more consistent than effort (whether considering number of boats, number of fishers, number of certain gear types, etc.[50–52]). Similarly, a jurisdiction's biomass status was defined relative to $B_{MMSY}$: whether or not its estimated biomass (weighted or not, t/km²) was below its jurisdiction-specific $B_{MMSY}$ estimates from the Gompertz-Fox model. A jurisdiction's fishing and biomass status was calculated as:

$$B_{status,c} = (B_c \, or \, B_{weighted,c}) / B_{MMSY,c} \tag{35}$$

$$F_{status,c} = C_c / MMSY_c \tag{36}$$

These are distributions, but in the main manuscript we report median and 0.9 quantiles. Finally, similarly to individual sites, for jurisdictions with both catch and biomass we calculated the percentage overfishing (i.e., catching above the maximum that can be sustained given their estimated biomass values) and the percentage in different fishery status categories (i.e., in good condition, warning, recovering or unsustainable) using the jurisdiction-specific surplus production curves and weighted biomass values.

## Trade-offs between production and ecosystem metrics

To assess the trade-offs between production and other ecosystem metrics on fished reefs, we evaluated the relationship between reef fish biomass and four ecosystem metrics thought to be important for ecosystem functioning[1,32]. Ecosystem metrics were: mean fish length (i.e., average observed length for species in the community; L; $n = 1763$), the probability of observing top predators (i.e., presence/absence (PA, $n = 1763$) of fish from the following families: Carcharhinidae, Ginglymostomatidae, Heterodontidae, Sphyrnidae, and Carangidae, Lutjanidae, Serranidae and Sphyraenidae above 50 cm), parrotfish scraping potential (P; $n = 1116$), and estimated total fish species richness (R; $n = 1753$). Note that sample sizes vary depending on whether or not that metric was provided by data providers at that scale. To account for species-abundance patterns and the effect that sampling area has on observed species richness, we estimated total fish species richness by fitting Poisson-lognormal distributions to the reef-scale observed species abundance distributions (i.e., counts of

individuals of different species), retaining those that fit well (99.8%) and estimating the fraction of total species richness revealed by the observed sample[66,67]. Potential parrotfish scraping rates (area grazed per minute) were calculated for sites including parrotfish as the product of parrotfish density, feeding rate and bite dimension (area[36]). Size specific-feeding rates and bite areas (mm$^2$) were taken from the literature[36,68,69], and supplemented with additional 3-min observations of species from the Red Sea and Indonesia[36]. As with biomass for fished reefs, methodological effects (Habitat type, Depth, Census Method and Sampling area), available environmental effects (Atoll, SST and ocean productivity) and human impact (i.e., total gravity) were accounted for using generalized multilevel models implemented in brms[70]. Model fits were examined (Supplementary Fig. 10) and model selection favored the full models for all metrics (Supplementary Table 2). We used a gaussian error family for log-transformed mean length and total species richness, a hurdle-lognormal family for parrotfish scraping potential (given the large number of zeros) and the Bernoulli family for presence/absence of top predators (with a logit link function):

$$\log(L_z) \sim N(I + \beta_1 x_{h_{flat,z}} + \beta_2 x_{h_{crest,z}} + \beta_3 x_{h_{backreef,z}} + \beta_4 x_{d_z} + \beta_5 x_{prod_z} + \beta_6 x_{cm_{ds,z}} + \beta_7 x_{cm_{pc,z}} + \beta_8 x_{sarea,z} + \beta_9 x_{atoll,z} + \beta_{10} x_{SST,z} + \beta_{11} x_{gr,z} + u_C, \sigma_L)$$

(37)

$$\log(R_z) \sim N(I_R + \beta_1 x_{h_{flat,z}} + \beta_2 x_{h_{crest,z}} + \beta_3 x_{h_{backreef,z}} + \beta_4 x_{d_z} + \beta_5 x_{prod_z} + \beta_6 x_{cm_{ds,z}} + \beta_7 x_{cm_{pc,z}} + \beta_8 x_{sarea,z} + \beta_9 x_{atoll,z} + \beta_{10} x_{SST,z} + \beta_{11} x_{gr,z} + u_C, \sigma_R)$$

(38)

$$if \, P_z = 0, P_z \sim bernoulli(\delta_P)$$

(39)

$$if \, P_z > 0, P_z \sim LN(I_P + \beta_1 x_{h_{flat,z}} + \beta_2 x_{h_{crest,z}} + \beta_3 x_{h_{backreef,z}} + \beta_4 x_{d_z} + \beta_5 x_{prod_z} + \beta_6 x_{cm_{ds,z}} + \beta_7 x_{cm_{pc,z}} + \beta_8 x_{sarea,z} + \beta_9 x_{atoll,z} + \beta_{10} x_{SST,z} + \beta_{11} x_{gr,z} + u_C, \sigma_P)$$

(40)

$$PA_z \sim bernoulli(\delta_{PA,z})$$

(41)

$$\text{logit}(\delta_{PA,z}) = I_{PA} + \beta_1 x_{h_{flat,z}} + \beta_2 x_{h_{crest,z}} + \beta_3 x_{h_{backreef,z}} + \beta_4 x_{d_z} + \beta_5 x_{prod_z} + \beta_6 x_{cm_{ds,z}} + \beta_7 x_{cm_{pc,z}} + \beta_8 x_{sarea,z} + \beta_9 x_{atoll,z} + \beta_{10} x_{SST,z} + \beta_{11} x_{gr,z} + u_C$$

(42)

where β… are the effect sizes for the covariates (estimated separately for each response variable), $I$… are the intercepts for the specific response variables, $u_c$ are the jurisdiction-level random effects, and δ… are the probabilities, probability of observing zero parrotfish scraping potential and probability of observing a top predator, respectively.

Next, to visualize and assess the potential trade-offs between production and ecosystem metrics we calculated the marginalized ecosystem metrics, corrected for both sampling and environmental effects, using "slopes", "standard belt transects", "non-atolls", and average sampling area, productivity, SST and depth as a reference. We did the same for biomass (for non-atolls and average environmental conditions) using the posterior effect sizes from the reference point model and compared the biomass gradient to these ecosystem metrics using generalized additive models (e.g., Fig. 3a). Note we marginalized for environmental conditions only for that component of the analyses, for the remaining analyses (e.g., Fig. 3b–i) we only marginalized for methodology given that a site is expected to have different production based on environmental conditions.

## Sensitivity analyses and additional model checks

**Accounting for the potential openness of reserve populations.** In contrast to previous work aiming to estimate baselines or reference points for coral reef fish, we wanted to take account of the possibility that reserves may export a portion of their biomass (and thus using reserve asymptotes as unfished biomass may bias reference points at the scale of metapopulation closure[20]). Consequently, we directly parameterized exports within our model (Supplementary Discussion 3): as a rate (biomass exported per biomass unit at each time step) or as a proportion of the community growth rate. Alternatively, we used the un-standardized gravity metric in our model, assuming high gravity locations would have greater net export of biomass due to the depletion of the surrounding seascape (mathematically, adjusting the biomass in reserves if human impact was above zero given the environmental and methodological variables accounted for in our model). Next, we compared these alternatives using model selection, to determine which of all the approaches performed best in terms of predictive accuracy (through leave-out one cross validation). Model selection favored the model including gravity (Supplementary Table 2). Finally, we compared our approach to the model that included exports as a proportion of the community growth rate but fixing exports at 0, 5, 10, 15, 20, 25, or 30%. Again, we performed model selection, and our full model (including gravity) was preferred (Supplementary Table 2). These analyses strongly suggest that our approach of making estimated reserve biomass a function of gravity adequately (albeit phenomenologically) captured potential effects of export on biomass dynamics within reserves. Additionally, when the export parameter was modeled explicitly as a rate or as a proportion of growth, this parameter was statistically non-identifiable (i.e., there was insufficient information in our data to estimate the parameter: see Supplementary Discussion 3 for details). This suggests that explicit estimation of the export parameter would likely require time series from a relatively large number of reserves distributed across a broad range of environmental conditions that prevail on reefs.

**Including vs. not including remote locations.** Remote reefs provide the best available data to estimate unfished biomass at the seascape scale once accounted for differences in environmental conditions. However, if these remote reefs differ from non-remote reefs in the data due to other unmeasured variables, and those variables promote higher biomass on remote reefs, estimates of unfished biomass for non-remote reefs could be biased upwards. To investigate whether unfished biomass estimates are biased upwards by the inclusion of remote reefs in our study, we ran the reference point model without including remote locations. When remote locations were not included, some parameters (e.g., r) were highly dependent on the priors used (i.e., not identifiable, posterior contraction of 0.34). Using the same prior unfished biomass as our main analyses, the median estimated unfished biomass for average environmental conditions was higher (~127 vs 116 t/km$^2$) and broader. Thus, we find no evidence that our inclusion of remote locations creates upward bias in our estimates of unfished biomass. Rather, they help to impose a realistic upper bound on unfished biomass.

**Potential MPA placement effects.** We used a space-for-time substitution among reserves of different ages to infer biomass recovery through time. This approach assumes that the biomass at reserve age zero ($B_{min}$) and the community growth rate of reef assemblages (r) do not vary among locations. These parameters likely vary among space (and time) and thus we mention the need for further reef fish compilations (e.g., time-series of multiple individual reserves) in our future directions section ("*Adapting reference points and assessments in the future*"). However, to make sure our parameters estimated from reserve data (e.g., $B_{min}$) are not biased for fished reefs we show that (i) there is substantial overlap in the distribution of environmental

covariate values among categories (Supplementary Fig. 4), (ii) fish species richness from high compliance marine reserve sites is within the distribution of exploited reef sites (Supplementary Fig. 14), and (iii) the estimated $B_{min}$ (initial biomass prior to reserve implementation) from the analysis of reserve dynamics is within the distribution of biomass estimates of openly fished sites (Supplementary Fig. 15).

**Choice of surplus production model.** Given that the shape of the surplus production function curve for community coral reef fish assemblages is not well known, we explored alternate special cases of the Pella-Tomlinson (P-T) surplus model[14]: a re-parametrized version of the Gompertz-Fox model[10,11] which allows MMSY to peak below 0.5 of unfished biomass, a Graham-Schaefer surplus[12,13] production model which allows MMSY to peak at 0.5 of unfished biomass, and two special cases of the P-T that allow production to peak at >0.5 of unfished biomass[71,72]. The P-T model has an extra parameter ($n$) that adjusts the standing stock biomass value at which production peaks. When $n = 2$ the P-T becomes the Graham-Schaefer and as $n \rightarrow 1$, the P-T approaches the Gompertz-Fox model. Before trying the different P-T versions, we first tried estimating the parameter $n$ by fitting the P-T model directly. However, the P-T model did not converge when we allowed $n$ to be estimated; probably because a range of "n" values could provide an equally good fit to our data, as has previously been noted in other contexts[12]. Consequently, we tested the different versions and compared them through leave-out-one cross-validation. Specifically, we compared the Gompertz-Fox (i.e., limit of P-T model as $n \rightarrow 1$), Graham-Schaefer (P-T with $n = 2$), and P-T models with $n = 3$ and $n = 4$[12].

Model selection favored the Gompertz-Fox model (Supplementary Table 2), yet the differences in the expected log predictive density values were small in magnitude, suggesting that additional reserve recovery data will help further discern the functional form of recovery. As a sensitivity analysis, we present details and results under different surplus models in Table 1 and the supplemental information (Supplementary Discussion 1). As expected, in comparison to the Gompertz-Fox Model, the Graham-Schaefer model and P-T versions with larger $n$ values provide larger $B_{MMSY}$ estimates; resulting in higher percentages of sites and jurisdictions classified as below $B_{MMSY}$ (Table 1). Nevertheless, relationships with environmental factors and MMSY estimates remain consistent among models (Supplementary Discussion 1).

**Choice of catch statistics.** We used spatially reconstructed catches from the SAUP intersected with tropical coral reef areas as a best estimate for site and jurisdiction-level reef fish catch per unit area, restricting fish to "reef associated fish" from the families included in our biomass data (Supplementary Table 1). However, catch estimates for coral reef fishes are uncertain, so we also repeated our Gompertz-Fox model fits using other catch estimates: (i) spatial reported, (ii) non-spatial reconstructed, (iii) non-spatial reconstructed excluding the industrial sector, (iv) non-spatial reported, and (v) non-spatial reported excluding the industrial sector. We performed (i) for site-level analyses (since the site-level analysis requires the use of the spatial catch data), and all of (i-v) for the jurisdiction-level analyses. For non-spatial data, as we did not have the geolocation of catches, we had to assume that reef associated fish from the families included in our analyses caught by a fishing entity were obtained from that jurisdictions' reef area. Note that by using non-spatial jurisdiction level data we were able to provide an estimate of the status of jurisdictions with reef area that returned NAs when intersected with coral reef area estimates, thus increasing the sample size (from 108 to 111). When only reported data is used, the percentage of sites catching above MMSY, overfishing and of conservation concern decreases by less than 10% —to 46%, 58% and 73%, respectively (in comparison to 56%, 65% and 77% when catch reconstructions are used). For jurisdictions, when reconstructed non-spatial catch data was used the percentage of jurisdictions catching above MMSY remained close to the spatial estimate of 45% (49% and

43%, for catch data either including or excluding the industrial sector, respectively), still yielding >50% (53% and 51%, respectively) of jurisdictions classified as "conservation concern". When only reported data was used (which is likely an underestimate of reef fish catch[23]), the percentage of jurisdictions catching above MMSY decreased to 30% when spatial data was used and to 34% or 28% when non-spatial reported data was used (including and excluding the industrial sector, respectively) but jurisdictions of "conservation concern" remained above 43%.

**Individual intrinsic growth rates vs. community biomass specific growth rates.** Here we estimated community biomass growth rates for coral reef-fish assemblages, which, to our knowledge, does not have a straightforward relationship between the average of individual species intrinsic growth rates. To show that community biomass growth rates do not necessarily represent an average of the species-specific intrinsic growth rates, but rather can fall at the high or low end of the distribution of those growth rates, we (i) provide the distribution of individual intrinsic growth rates estimated from FishLife[73] for our reference point reserve and remote data (to the lowest taxonomic level possible) and compare it to our estimates of community growth rate from the Gompertz-Fox model; and (ii) using the fish communities in ref. 1, we also show how for those communities, relating community growth rates to species intrinsic growth rates is not straightforward (Supplementary Discussion 4). Differences between community and species-specific biomass growth rates could arise from shifts in the contribution of different species to overall biomass growth at different stages in the community biomass recovery process, due, for example, to species interactions, or to slower recovery of larger slower-growing groups that increase in relative abundance as community biomass increases (e.g.,[74]). Thus, we outline this area as a future direction to be explored, especially in relation to understanding how multispecies reference points should be adapted if systems transition to alternate stable states that differ substantially in species compositions.

**Reporting summary**

Further information on research design is available in the Nature Portfolio Reporting Summary linked to this article.

## Data availability

For the main analyses of this study we compiled several existing datasets. We compiled three datasets on underwater reef associated fish and associated data[16,17,32]. These used published species-specific length-weight relationships available from FishBase (http://fishbase.org) to calculate reef fish biomass. Reconstructed reef fish catch estimates (in metric tonnes) were obtained from the Sea Around Us Project (SAUP) catch database (http://www.seaaroundus.org). We also used the tropical coral reef spatial grid (https://data.unep-wcmc.org/datasets/1) to intersect with catch data and obtain site-specific and jurisdiction level reef fish catches. Additionally, several site-specific covariates in our model were obtained from online data sources: human impact (https://research.jcu.edu.au/data/published/a9167f52dba39f693f55ae68a0a5dccf/), sea surface temperature (https://coralreefwatch.noaa.gov/) and ocean productivity (https://sites.science.oregonstate.edu/ocean.productivity/). Data used for this paper is available as Supplementary Data: Supplementary Data 1 contains reef site-scale data used in the main analyses. Supplementary Data 2 contained jurisdiction-scale data used in the main analyses. Supplementary Data 3 contains individual fish specific data used to estimate species richness and perform some sensitivity analyses.

## Code availability

Code used for this paper is available from GitHub (https://github.com/JZamborain-Mason/ZamborainMasonetal_2023_ReefSustainability;[75]).

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

## Acknowledgements

The authors would like to thank Andreas Dietzel for providing constructive comments on reef area estimates, Eva Maire for providing travel time estimations, Gordon Tsui for providing catch data cell ID's, Michel Kulbicki and Tim McClanahan for providing data and constructive comments on the manuscript, and Pascale Chabanet, Eran Brokovich, Marah Hardt, Juan Cruz, Laurent Vigliola, Mark Tupper, and Stuart Sandin for providing data. We also thank all the data collectors. JZM, SRC, AH and JEC thank the Australian Research Council for funding support. MAM was supported by an NSERC Canada Research Chair.

## Author contributions

J.Z.M. conceived the study with supervisory support from S.R.C., J.E.C., and M.A.M.; J.Z.M. developed and implemented the analyses with support from S.R.C., J.E.C., M.A.M., and N.G. A.H. calculated the parrotfish scraping potential. N.G., J.E.C., M.A.M., A.H., M.G., A.B., D.B., G.E., D.F., S.F., A.F., C.C., A.G., D.M., N.P., R.S.S., L.W., I.W., and S.W. contributed data. J.Z.M. wrote the manuscript with help from S.R.C., J.E.C., and M.A.M. All remaining authors (N.G., A.H., M.G., A.B., D.B., G.E., D.F., S.F., A.F., C.C., A.G., D.M., N.P., R.S.S., L.W., I.W., and S.W.) made substantive contributions to the text.

## Competing interests

The authors declare no competing interests.

## Additional information

[1]Harvard T.H. Chan School of Public Health, Boston, MA 02115, USA. [2]College of Science and Engineering, James Cook University, Townsville, QLD, Australia. [3]ARC Centre of Excellence for Coral Reef Studies, James Cook University, Townsville, QLD, Australia. [4]Ocean Frontier Institute, Department of Biology, Dalhousie University, Halifax, NS B3H 3J5, Canada. [5]Lancaster Environment Centre, Lancaster University, Lancaster LA1 4YQ, UK. [6]School of Biology, Faculty

of Biological Sciences, University of Leeds, Leeds, West Yorkshire LS2 9JT, UK. [7]Centre for Biodiversity and Conservation Science, School of Biological Sciences, University of Queensland, Brisbane, Queensland 4072, Australia. [8]Coastal Research Center, Marine Science Institute, University of California, Santa Barbara, CA 93106, USA. [9]School of Life Sciences, University of Technology Sydney 2007 Australia, Ultimo, Australia. [10]Institute for Marine and Antarctic Studies, University of Tasmania, Hobart TAS 7001, Australia. [11]MRAG Ltd, 18 Queen Street, London W1J 5PN, UK. [12]Leibniz Centre for Tropical Marine Research (ZMT), 28359 Bremen, Germany. [13]Faculty of Biology & Chemistry (FB2), University of Bremen, 28359 Bremen, Germany. [14]National Geographic Society, Pristine Seas Program, 1145 17th Street N.W, Washington DC 20036-4688, USA. [15]Hawaiʻi Institute of Marine Biology, Kāneʻohe, Hawaiʻi 96744, USA. [16]Blue Ventures, The Old Library, Trinity Road, Bristol BS2 0NW, UK. [17]King Abdullah University of Science and Technology, Thuwal, Saudi Arabia. [18]MARBEC, Univ Montpellier, CNRS, Ifremer, IRD, Montpellier, France. [19]School of Natural & Environmental Sciences, Newcastle University NE17RU, Newcastle upon Tyne, UK. [20]University of New Caledonia, BPR4 98851, Noumea cedex, New Caledonia. [21]Coral Reef Ecosystems Division, NOAA Pacific Islands Fisheries Science Center, Honolulu, HI 96818, USA. [22]Oceans Institute, University of Western Australia, Crawley, WA 6009, Australia. [23]Department of Biodiversity, Conservation and Attractions, Kensington, Perth, WA 6151, Australia. [24]Smithsonian Tropical Research Institute, Panama City, Panama. ✉e-mail: jzm@hsph.harvard.edu

