## [Peer Review File · Nature Communications]

Sustainable reference points for multispecies coral reef fisheriesREVIEWER COMMENTS

Reviewer #1 (Remarks to the Author):

Review of Sustainable reference points for multispecies coral reef fisheries

This paper uses an extensive data set of reef biomasses, environmental covariates and reconstructed catches to determine broad and jurisdictionally specific reference points for coral reef fisheries – MMSY and BMMSY - and then gives a simple assessment of status of those fisheries. Space for time substitution (e.g. use of remote reefs as a proxy) provides a way of realistically anchoring the results. The work is extensive and well thought through with the methods well presented in the main manuscript and supporting supplementary information. The code also seems well laid out and commented meaning it is straightforward to follow.

In a few spots assumptions require a little more justification and it would be nice to see the implications of uncertainty presented in the main paper rather than all in the supplementary materials. There are also a few minor typographical or formatting issues to deal with.

In summary though an impressive piece of work that only requires minor revision before publication in my assessment. Please see specific comments in the attached files.

Reviewer #2 (Remarks to the Author):

This paper assembles a large data set on coral reef fisheries abundance and estimates the status of the fisheries in these reefs. The methods appear to be appropriate and overall the paper is very well done. The use of a multispecies production function and evaluation in relation to PGMMSY was a good choice.

My only concerns are associated with the jurisdictional analysis, see my detailed comments below. Generally it isn't clear how the authors have moved from a series of sites in a jurisdiction where there are abundance surveys, and merge this with what is generally a country level estimate of catch.

Line 101: define what measures of "local environmental conditions" were used earlier in introduction

Line 112 "estimate" instead of "expected"?

Lines 154-156. The analysis at the jurisdiction level seems unclear. My reading is that for those jurisdictions you have estimates of catch, and total reef area, but there is not data on how much reef area is associated with the different covariates such as coral cover etc., and I don't see an explanation of how the total production potential for a jurisdiction is calculated. The text states that there are jurisdictions where both catch and biomass are available. That needs more explanation, I find it difficult to believe that many jurisdictions are completely surveyed, so there must be some assumptions involved. Also it would be helpful to have an explicit look at sites where biomass and catch were available on a site specific, rather than jurisdictional basis.

Lines 164-166 it would be better and more consistent to report what fraction of sites are below the level that produces PGMMSY rather than below BMSY, because areas only slightly below BMSY are losing little if any yield and fluctuations above and below BMSY are expected in well managed fisheries.

Line 389: There have been concerns expressed about the SAUP reconstructions of catch in some countries. It would be useful for the taxa involved to show how much the SAUP estimates differ from the landings data reported to FAO.

Ray Hilborn

Response to referees

We thank the editor and the referees for their time, understanding and constructive suggestions for further improvement of our manuscript.

Below we provide the response to the referees' comments. Original comments are in black, our responses are in blue. References to the manuscript (with track changes) are detailed, if appropriate, in each section.

Note that we have copied, pasted and addressed the minor comments that Referee #1 (i.e., Reviewer #1) made throughout the main document and supplementary information at the end of this response.

REVIEWER COMMENTS

Reviewer #1 (Remarks to the Author):

Review of Sustainable reference points for multispecies coral reef fisheries

This paper uses an extensive data set of reef biomasses, environmental covariates and reconstructed catches to determine broad and jurisdictionally specific reference points for coral reef fisheries – MMSY and BMMSY - and then gives a simple assessment of status of those fisheries. Space for time substitution (e.g. use of remote reefs as a Bo proxy) provides a way of realistically anchoring the results. The work is extensive and well thought through with the methods well presented in the main manuscript and supporting supplementary information. The code also seems well laid out and commented meaning it is straightforward to follow. We thank the referee for their rigorous evaluation and support for our study.

In a few spots assumptions require a little more justification and it would be nice to see the implications of uncertainty presented in the main paper rather than all in the supplementary materials. There are also a few minor typographical or formatting issues to deal with. We thank the referee for specifying the need for justification. We have now addressed all the minor points the referee highlighted in the text (copied and addressed below). To address the referee's uncertainty point, we have now added to the main text relevant results from the different surplus production models that were previously in the supplementary information (Table 1 and the associated text).

In summary though an impressive piece of work that only requires minor revision before publication in my assessment. Please see specific comments in the attached files. Again, we thank the referee for their support and constructive feedback. At the end of this document, we have copied, pasted and addressed the specific comments that the referee made throughout the main document and supplementary information.

Reviewer #2 (Remarks to the Author):

This paper assembles a large data set on coral reef fisheries abundance and estimates the status of the fisheries in these reefs. The methods appear to be appropriate and overall the paper is very well done. The use of a multispecies production function and evaluation in relation to PGMMSY was a good choice.

We thank the referee for their constructive comments and overall support for our work.

My only concerns are associated with the jurisdictional analysis, see my detailed comments below. Generally it isn't clear how the authors have moved from a series of sites in a jurisdiction where there are abundance surveys, and merge this with what is generally a country level estimate of catch.

We thank the referee for highlighting the need for clarification. In summary, following the referee's detailed comment below (i.e., "Lines 154-156" below and associated response), we have now (i) also performed analyses at a site-scale, and (ii) explicitly stated our jurisdiction-level methods and assumptions in the main text: "*At the jurisdiction scale, we grouped expected biomass (also weighted by the proportion of marine protected areas in a jurisdiction; n=49; Methods) and per-unit-area catch estimates (n=111) and compared these to jurisdiction-specific sustainable reference points (Fig. 2a-b; Supplementary Fig. 5), assuming that our sampled reefs were representative of jurisdiction-level conditions (Methods).*" (line 168-with track changes).

Line 101: define what measures of "local environmental conditions" were used earlier in introduction

We thank the referee for specifying the need for clarification. We have addressed the referee's comment by adding "*(i.e., , reference points are estimated as explicit functions of sea surface temperature, ocean productivity, hard coral cover and whether the reef is an atoll)*" (line 105).

Line 112 "estimate" instead of "expected"?

Accepted change (line 119).

Lines 154-156. The analysis at the jurisdiction level seems unclear. My reading is that for those jurisdictions you have estimates of catch, and total reef area, but there is not data on how much reef area is associated with the different covariates such as coral cover etc., and I don't see an explanation of how the total production potential for a jurisdiction is calculated. The text states that there are jurisdictions where both catch and biomass are available. That needs more explanation, I find it difficult to believe that many jurisdictions are completely surveyed, so there must be some assumptions involved. Also it would be helpful to have an explicit look at sites where biomass and catch were available on a site specific, rather than jurisdictional basis. We thank the referee for specifying the need for clarification. The referee is correct in that the jurisdictions were not completely surveyed and, given data availability, we had to assume that the distribution of sampled reefs within a jurisdiction (and their associated uncertainty) were a representative measure of jurisdiction-level biomass and environmental conditions (e.g., coral cover) after accounting for potential methodological and habitat differences. These total production calculations for jurisdictions can be found in the methods (e.g., line 712) and an analysis showing that our sampled sites are representative with respect to their jurisdictions in terms of human impact (a good proxy for biomass; refs 21,32) can be found in former Fig. SI1-8

(i.e., Supplementary Fig. 13). However, to more fully address the referee's comment and add clarity, we have now explicitly stated our jurisdiction-level methods (and assumptions) in the main text. More specifically, we have written : *"At the jurisdiction scale, we grouped expected biomass (also weighted by the proportion of marine protected areas in a jurisdiction; n=49; Methods) and per-unit-area catch estimates (n=111) and compared these to jurisdiction-specific sustainable reference points (Fig. 2a-b; Supplementary Fig. 5), assuming that our sampled reefs were representative of jurisdiction-level conditions (Methods)." (line 168). Note that we also highlight the future data needed to tackle this assumption in the "Adapting reference points and assessments in the future" section (e.g., "Here we provided fishery assessments at site and jurisdiction scales using available catch and biomass statistics, providing a global overview of the status of reef fish stocks. However, we acknowledge that uncertainty about stock status, catch statistics and their geolocation, and spatio-temporal heterogeneity within jurisdictions means that improved precision of estimates at the spatial and temporal scales appropriate to management are needed to better inform decisions by resource practitioners"³⁸ - line 359). Additionally, following the referee's suggestion, we have now expanded our site-level analyses, incorporating site-level fishing and fishery status estimates using site-level spatial catch statistics. We find that, based on reconstructed catch statistics, 65% of our sites are experiencing overfishing, with 56% having catch estimates above MMSY. This implies 77 % of sampled sites open to extraction have stocks of conservation concern, having passed one or both MMSY reference points. This analysis further strengthens our main conclusions and has been included throughout the main text (e.g., "56 [49-63] % of sites had catch per-unit-area estimates above MMSY"-line 186), and methods (e.g., "Then for reefs open to extraction, we compared these site-specific marginalized biomass estimates ($B_{(marg,z)}$) and per-unit-area catch estimates (C_z) to their estimated B_{MMSY} and MMSY values, defining a location as to whether its biomass status ($B_{(status,z)}$) or fishing status ($F_{(status,z)}$) were below or above/equal to 1."-line 676).*

Lines 164-166 it would be better and more consistent to report what fraction of sites are below the level that produces PGYMSY rather than below BMSY, because areas only slightly below BMSY are losing little if any yield and fluctuations above and below BMSY are expected in well managed fisheries. We thank the referee for providing this suggestion. We believe both reference points are useful, especially considering all or the majority of these fisheries are currently not managed for MMSY. Thus, to address the referee's comment we now report both the number of sites below BMMSY and below the (lower-bound) biomass level that produces PGMSY. For example, in the main text we have added: *"23 % of sites were below the lowest biomass value that produces PGMY, and less than half (i.e., 47%) were in the biomass range of producing PGMY (i.e., estimated to be producing at least 80% of their maximum sustainable catch potential)" (line 191).*

Line 389: There have been concerns expressed about the SAUP reconstructions of catch in some countries. It would be useful for the taxa involved to show how much the SAUP estimates differ from the landings data reported to FAO.

We agree with the referee and thank them for bringing this up. Acknowledging the referee's point, in our analyses we perform a sensitivity analysis to the choice of catch statistics (e.g., line line 934). More specifically, two of the sensitivity checks consisted in re-running all our analyses

using only reported landings (spatial and non-spatial). Our results (extended in the Methods) show how the percentage of sites and jurisdictions (optimistically assuming protected regions are at unfished conditions) classified as conservation concern change from 77 and 53 % to 73% and 42 %, respectively, if we only use reported catches. Details of the sensitivity analysis were in the methods. However, to address the referee's point we have now detailed the results in the main text (i.e., *"Furthermore, assessment results did not improve substantially when we used reported catches instead of catch reconstructions (Methods); sites and jurisdictions of "conservation concern" decrease to 73 and 42%, respectively, indicating that our findings are not substantially driven by biases in the catch reconstruction methodology."*; line 257).

Reviewer #1 Specific comments within the documents

Here, we have copied, pasted and addressed the minor comments that referee 1 made throughout the document and supplementary information:

Main text

line 95: replace such with that Accepted change.

line 96: replace under by generated using Accepted change.

line 107-103: "I think more could be made of this in the discussion. Care to comment on using those numbers in anger?" We thank the referee for their suggestion. The estimated variability in site-specific reference points is discussed throughout the manuscript (e.g., *"and highlights the importance of accounting for local context when assigning fisheries reference points"*; line 124-with track changes). However, to address the referee's comment we have now expanded our discussion section to clarify the importance of accounting for such variability i.e., *"As opposed to previous work^(e.g., 16,17), we show that reference points can vary greatly among locations given their local context, and such variability can have materially different implications for local fisheries management"*-line 356), acknowledging that it is typically not accounted for (e.g., refs 16,17). We are not sure what the referee suggests about using those numbers in "anger" but presumably there is a tone to what we have written that was suggestive. We have tried to remain neutral in our writeup so if there is something specific we can address we would be open to hearing about it.

line 115; add "and" Accepted change.

line 116: replace "and" for "It also" Accepted change.

line 116: delete "for example" Accepted change.

line 294: "How difficult would it be to represent more than the Gompertz-Fox? So highest and lowest outcomes etc?" We thank the referee for this suggestion. To address the referee's comment, we now explicitly state the range of values under different surplus production models in a table in the main text (i.e., Table 1).

line 315-319: Might want to refer to Fulton et al here Done! Thank you for this reference.

line 369: Explain why. We thank the referee for specifying the need for clarification. We have addressed the referee's comment by adding *"This was done because the majority of sites were only sampled once"* (line 436).

line 394: Given the uncertainty of that dataset did you look at a range of “mean total catch” - basically how did you propagate uncertainty? While we had originally provided a sensitivity analysis to the choice of catch statistics (e.g., spatial vs non-spatial; reconstructed vs reported...), we had not tested different yearly spatial catch data reconstructions, using the mean total catch on reefs per fishing entity (from years 2008 to 2014). However, to address the referee's point (and the referee's point below-line 399), we have now (i) replaced spatial catch estimates by the “catch over reefs divided by reef area” (instead of catch over reefs per fishing entity divided by their reef area); and (ii) re-run our analyses using individual year spatial catch besides the mean. We find that (i) individual year spatial catch reconstructions are highly correlated with each other providing identical status results in comparison to using the mean (e.g., >0.99 Pearson's correlation between the mean and individual years; Fig. below), which also means that using yearly catch statistics to represent uncertainty in catches would likely underestimate true catch uncertainty; and (ii) using the proposed estimate of spatial catch (instead of using “fishing entity”) does not change our study results or conclusions. This is now clarified in the methods: (i) “At the jurisdiction-scale, we calculated the catch per unit area (catch/km²/y) by dividing a jurisdiction's estimated mean total reef fish catch that overlapped with global reef polygons by the estimated total jurisdiction reef area²⁵” (line 574); “Note that we used the mean catch because we only used seven points in time, highly correlated with each other (Pearson's correlation coefficient > 0.99)” -line 481.

Fig Pairs plot showing the correlation between using the mean spatial catch or individual years. Each point is a jurisdiction. Line is the unity line. Correlations are Pearson's coefficients.

line 396: why We thank the referee for their comment. This comment is no longer relevant when addressing the referee's comment below (i.e., line 399). In other words, when not using the fishing entity's catch as a unit of analysis, we no longer have catches from outside jurisdictions.

line 399: "Could this assumption get you into trouble - either globally or in specific regions (e.g. SE Asia)?" We thank the referee for pointing this out, which made us re-do our spatial catch analyses using jurisdiction catches instead of fishing entity catches. When we did not have the geolocation of catches (e.g., for the non-spatial data), we had to assume reef associated fish from the families included in our analyses caught by a fishing entity were caught over their reef area. However, when we did have an estimate of the geolocation of the data (i.e., spatial catch data), we could intersect catch reconstructions with coral reef polygons and estimate a jurisdiction's fishing status by comparing the catches on the jurisdiction's reefs relative to the jurisdiction's reef area. To address the reviewer's comment, we have now (i) disregarded fishing entity in the main spatial analyses, (ii) added clarity and specified the reasoning of our fishing entity assumption in the text when we use non-spatial data (e.g., "For non-spatial data, as we did not have the geolocation of catches, we had to assume reef associated fish from the families included in our analyses caught by a fishing entity were obtained from that jurisdictions' reef area"-line 943), and (iii) stated the catch data uncertainty more clearly in the future directions section (i.e., line 359: "we acknowledge that uncertainty about stock status, catch statistics and their geolocation, and spatio-temporal heterogeneity within jurisdictions means that improved precision of estimates at the spatial and temporal scales appropriate to management are needed to better inform decisions by resource practitioners³⁸").

line 404: replace the biomass of " for "in terms of biomass" Accepted change.

line 412: Did you do any standardisation or "corrections" to account for catchability, gear use, effort levels/type etc through time? We thank the referee for specifying the need for clarification. We did not, as we are analyzing fisheries independent biomass data (e.g., we are not using catch rates as a proxy for abundance). Therefore, we consider catch potential and/or ecological availability of the multispecies assemblage irrespective of method of capture and catchability. We have clarified this point in the methods: "(i.e., catch potential and/or ecological availability of the multispecies assemblage irrespective of method of capture and catchability)" (line 495).

line 414: delete "although they" Accepted change.

line 415: add "Nevertheless, they do" Accepted change.

line 428: You might want to make clear that this is of the entire community Done. We have added "community" before biomass (line 512).

line 583: Any reefs with sufficient data to check some of these assumptions hold? Sorry if we are misunderstanding but what assumption is the referee wanting us to check here? The referee added this comment to "Then for reefs open to extraction, we compared these marginalized biomass estimates to their estimated B_{MMSY} value, defining a location as to whether its biomass status ($B_{status,z}$) was below or above/equal to 1". We do not assume anything here, just define a place as below or above their estimated B_{MMSY} . Assuming the referee is asking if we have sites that have temporal data to see if status estimates provide consistent trends, we do not currently, although much of this work is underway.

line 592: Doesn't MMSY give you C_{pot} by definition? What am I missing? We thank the referee for highlighting the need for clarification. C_{pot} is the potential sustainable yield of a given location relative to MMSY based on its CURRENT biomass. In other words, if a location is not at B_{MMSY} , the sustainable catch of that location is not MMSY but lower. C_{pot} would be equal to 1 if a location was at B_{MMSY} and lower if standing biomass was

above or below. We have now clarified this in the text. More specifically, we have rephrased the section to read as *“To estimate the relative catch potential for our sites, we also calculated the potential sustainable yield or surplus (PZ) of that site, conditional on its estimated biomass, and we expressed this relative to that site’s estimated MMSY (MMSYz), which is of course the catch potential of a site for the specific case when the estimated biomass of the site is equal to BMMSY:....[EQUATIONS]..... C_(pot,z) would thus have a value of one if a location’s biomass (Bmarg,z)) is at BMMSY,z and below one as the biomass is above or below BMMSY,z..”* (line 688).

line 613: deleted space Accepted change.

line 631: change of for or Accepted change.

line 640: Which two scenarios? Thank you for specifying the need for clarification. We have now rephrased the sentence to : *“It is this optimistic scenario that is shown in our figure (Fig. 2), although eight jurisdictions (PRIA, Australia, Hawaii, Belize, Reunion, New Caledonia, Mexico and Northern Mariana Islands) changed status if we used only the rees open to extraction (from above to below B_{MMSY}; Extended Data Fig. 6).”* (line 753)

line 658: change e.g., for “whether considering” Accepted change.

line 677: add “t” Accepted change.

line 737: change “on biomass (1/t” for “(biomass exported per biomass unit at each time step” Accepted change.

line 779: Was this expected? Or are there some high biodiversity fished reefs out there just because they were insanely rich in the first place or they haven’t been fished for long or are maintaining their biodiversity some other way We thank the referee for their comment.

We compared the species richness distributions in reserve and fished sites to make sure our inferences from sampled reserve sites were not biased by potential reserve placement effects. Answering the referee’s question, yes, we do expect some fished reefs to have high species richness, for all reasons the referee stated. As this does not change anything in our analyses, we have not made any changes to our text in relation to this comment.

Line 805: What does fairly consistent mean? We thank the reviewer for the need for specificity. We have deleted “fairly”.

line 833: This might be fine, just checking you’re not supposed to use the name of the author etc. We thank the referee for highlighting this. References for this journal need to be numbered and in superscript. Thus, we have not made any changes with respect to this comment.

line 833: add “also” Accepted change.

line 834: replace “it is also not straightforward how” for “relating” Accepted change.

line 834: delete “relate” Accepted change.

line 835: add “it is not straightforward” Accepted change.

Supplementary information 1

line 61: While I appreciate model fit is given in Extended Data Table 2 it would still be nice to see plots of these fits vs data We now provide the model fits for the different surplus production models in the supplementary information (Supplementary Figure 18).

line 68: Can you jitter so you can see alignment across models instead of single answer per covariate? We now provide the model coefficients jittered (Supplementary Figure 17).

line 113,114,181,190: *Not how you formatted Ref1 so just checking on consistency; Formatting?* We thank the referee for picking this up. We have now reformatted the references in accordance with journal guidelines.

line 118: delete "had" and add "ed" Accepted change.

line 146: replace "such" for "the" Accepted change.

line 183: delete "it is not straightforward" Accepted change.

line 185: add "is not straightforward" Accepted change.

line 221: *I'm assuming the ordering of sites in (a) is in terms of length of closure? What is effect of Australia or Peros Banhos outlines on the shape?* We think the referee meant to add this to former figure caption "Fig. SI1.6" (line 227; Supplementary Fig. 12), not former "Fig. SI1.5" (Supplementary Fig. 11), which is unrelated to the comment. Answering the referee's questions: (i) The ordering of sites is by mean travel time to human settlements for those sites. We have now added this in the figure caption: "*The y axis is ordered by mean travel time to human settlements.*" (line 236); (ii) taking away Australia or Peros Banhos does not change our conclusion that we should use 20h away from human settlements instead of 10h- in fact, it would reinforce our conclusion (e.g., Australia's sampled reefs have higher biomass than Palmyra Atoll, so taking Australia out would still show that reefs <20h away from human settlements have less biomass than those >20h of human settlements). As a consequence, we have not made any changes with respect to this comment.

Supplementary information 2

line 4,82,87,101,166,185: *Formatting?* We thank the referee for picking this up. We have now reformatted the references in accordance with journal guidelines.

line 24: *Why not cases with only some of the covariates?* We compare our full model to a null model to test support for our included covariates. We included covariates that theoretically made sense based on previous work, and did not want to data-mine our analyses by testing all the potential multiple different combinations of covariates (which would also require increased computational time). We have not made any changes with respect to this comment.

REVIEWERS' COMMENTS

Reviewer #1 (Remarks to the Author):

This paper presents work on coral reefs, drawing on a large multi-country dataset to provide global and site specific reference points for sustainable extraction and then checking the status of individual reefs vs those reference points.

The revision address all previous points and should be accepted for publication. There are a few minor typos to correct as noted below

1. Line 237: This should be "...and compared them to jurisdiction-specific..."
2. Line 363: It should be "below" not "bellow"
3. Line 369-370: I think this should be "... based on available catch statistics."
4. Lines 411-413, while you can infer sites vs jurisdictions in these sentences likely still best to be explicit that x% of sites or y% of jurisdictions were
5. Lines 415-420: An alternative wording that might be easier on the reader is:

When using the other surplus production models, all of which yield larger BMMSY reference points (Table 1), the percentage of locations classified as "conservation concern" increases from 77% of sites 85% and from 53% of jurisdictions up to 71 %. Furthermore, assessment results did not improve substantially when we used reported catches instead of catch reconstructions (Methods); with sites of "conservation concern" decreasing to 73% and jurisdictions to 42%.

Reviewer #2 (Remarks to the Author):

The authors have responded very well to the issues I raised on the initial review and I am happy with those changes and it is an excellent paper.

The only issue I would raise, and I should have done this in the initial review, is equating being below a management reference point and being classified as "conservation concern." Remembering that MSY or MMSY reference points are predicated on maximizing (or nearly so) sustainable yield, this is a far different standard than conservation concern associated with IUCN listing.

For a fishery to have $F > F_{MSY}$ does not imply either risk of extinction, or even continued decline in abundance, simply loss of some potential yield.

In the current use of the phrase "conservation concern" many if not most readers equate conservation concern with risk of extinction or collapse, rather than with failing to maximize yield.

So I would recommend either reserving the term "conservation concern" for seriously depleted stocks, or making it very clear that the use of the term does not imply either risk of extinction or expectation of future collapse.

Response to referees

We thank the editor and the referees for their time, understanding and suggestions for further improvement of our manuscript.

Below we provide the response to the referees' comments. Original comments are in black, our responses are in blue. References to the manuscript (with track changes) are detailed, if appropriate, in each section.

REVIEWER COMMENTS

Reviewer #1 (Remarks to the Author):

This paper presents work on coral reefs, drawing on a large multi-country dataset to provide global and site-specific reference points for sustainable extraction and then checking the status of individual reefs vs those reference points.

The revision address all previous points and should be accepted for publication. There are a few minor typos to correct as noted below

We thank the referee for their constructive comments and support. We are glad our revisions addressed the referee's points.

1. Line 237: This should be "...and compared them to jurisdiction-specific..." We thank the reviewer for noticing this typo. We have deleted "these" (line 158).

2. Line 363: It should be "below" not "bellow". We thank the reviewer for noticing this typo. We have deleted the "l" (line 189).

3. Line 369-370: I think this should be "... based on available catch statistics." ..." We thank the reviewer for noticing this typo. We have deleted "catch" (line 195).

4. Lines 411-413, while you can infer sites vs jurisdictions in these sentences likely still best to be explicit that x% of sites or y% of jurisdictions were Following the referee's comment, we have now been explicit about the assessment resolution (lines 225-229).

5. Lines 415-420: An alternative wording that might be easier on the reader is: When using the other surplus production models, all of which yield larger BMMSY reference points (Table 1), the percentage of locations classified as "conservation concern" increases from 77% of sites 85% and from 53% of jurisdictions up to 71 %. Furthermore, assessment results did not improve substantially when we used reported catches instead of catch reconstructions (Methods); with sites of "conservation concern" decreasing to 73% and jurisdictions to 42%. We thank the referee for suggesting this framing. We have now adopted it with very minor changes (lines 231-241).

Reviewer #2 (Remarks to the Author):

The authors have responded very well to the issues I raised on the initial review and I am happy with those changes and it is an excellent paper.

We thank the referee for their constructive comments and support. We are glad our revisions addressed the referee's previous concerns.

The only issue I would raise, and I should have done this in the initial review, is equating being below a management reference point and being classified as "conservation concern."

Remembering that MSY or MMSY reference points are predicated on maximizing (or nearly so) sustainable yield, this is a far different standard than conservation concern associated with IUCN listing.

For a fishery to have $F > F_{MMSY}$ does not imply either risk of extinction, or even continued decline in abundance, simply loss of some potential yield.

In the current use of the phrase "conservation concern" many if not most readers equate conservation concern with risk of extinction or collapse, rather than with failing to maximize yield.

So I would recommend either reserving the term "conservation concern" for seriously depleted stocks, or making it very clear that the use of the term does not imply either risk of extinction or expectation of future collapse.

We thank the referee for specifying the need for clarification. We classified stocks of conservation concern following a similar terminology to Costello et al. 2016 ("Global fishery prospects under contrasting management regimes). However, we totally understand the reviewer's concern and have clarified their point in the main text and methods. More specifically, we have added: "i.e., $C > MMSY$ and/or $B < B_{MMSY}$;" (line 223) and "Finally, similar to reference 4, we classified locations as "conservation concern" when catch was above MMSY and/or biomass was below B_{MMSY} . Such "conservation concern" status does not imply risk of extinction or expectation of collapse; instead, it implies fisheries management is likely needed to restrict catches and/or recover reef fish stocks to maximize long-term fisheries production." (lines 659-663).